# Remapping parasite landscapes: Nationwide prevalence, intensity and risk factors of schistosomiasis and soil-transmitted helminthiasis in Rwanda

Ladislas Nshimiyimana[1]ᵒ*, Jean Bosco Mbonigaba[1]ᵒ*, Aimable Mbituyumuremyi[1], Alison Ower[2], Dieudonne Hakizimana[3]*, Elias Nyandwi[4], Karen Palacio[2], Alphonse Mutabazi[1], Jeanne Uwizeyimana[5], Leonard Uwayezu[6], Michee Kabera[1], Emmanuel Hakizimana[1], Phocas Mazimpaka[1], Emmanuel Ruzindana[7], Eliah Shema[8], Tharcisse Munyaneza[7], Jean Bosco Mucaca[7], Maurice Twahirwa[7], Esperance Umumararungu[7], Joseph Kagabo[8], Richard Habimana[8], Elias Niyituma[1], Tonya Huston[5], Jamie Tallant[2], Warren Lancaster[2], Nadine Rujeni[8], Eugene Ruberanziza[2]

1 Division of Malaria and Other Parasitic Diseases, Rwanda Biomedical Centre, Ministry of Health, Kigali, Rwanda, 2 The END Fund, New York, United States of America, 3 Department of Global Health, University of Washington, Seattle, Washington, United States of America, 4 Centre for Geographic Information Systems and Remote Sensing (CGIS - UR), College of Sciences and Technology, University of Rwanda, Kigali, Rwanda, 5 Heart and Sole Africa, Kigali, Rwanda, 6 Byumba Level Two Teaching Hospital, Gicumbi, Rwanda, 7 National Reference Laboratory (NRL) Division, Rwanda Biomedical Centre, Ministry of Health, Kigali, Rwanda, 8 School of Health Sciences, College of Medicine and Health Sciences, University of Rwanda, Kigali, Rwanda

ᵒ Contributed equally as the first authors
* ladi8n@gmail.com (LN); ddhakizimana@gmail.com (DH)

## Abstract

### Background

Soil-transmitted helminthiasis (STH) and schistosomiasis (SCH) remain significant public health challenges in Rwanda, affecting individuals across all age groups. Despite ongoing mass drug administration (MDA) efforts, updated data on prevalence and risk factors are crucial for effective control and elimination strategies. This study reassessed the prevalence of STH and SCH in both children and adults in Rwanda, along with their associated risk factors, to guide control efforts.

### Methodology

A nationwide survey was conducted across 30 districts, testing 17,360 individuals for STH and 17,342 for *Schistosoma mansoni* using Kato-Katz (KK) and Point-Of-Care Circulating Cathodic Antigen (POC-CCA) tests. Mixed-effects logistic regression models were used to identify risk factors while accounting for district-level variability.

**Data availability statement:** All data supporting the findings of this study are included in the article. The dataset is held by the Rwanda Biomedical Centre and cannot be publicly shared due to restrictions imposed by the Rwanda National Ethics Committee. Participant consent did not permit sharing data beyond the research team, and public disclosure would violate these terms. However, access to the data may be granted upon request, pending approval from the Rwanda Biomedical Centre and the Rwanda National Ethics Committee. To request access, please contact the Rwanda Biomedical Centre at info@rbc.gov.rw, providing a detailed rationale for your request.

**Funding:** This work did not receive specific grant funding allocated directly to the authors. Funding to support this work was disbursed by the END Fund to the Rwanda Biomedical Centre (RBC) as part of the longstanding partnership under the program titled "Prevention, Control and Surveillance of NTDs in the Republic of Rwanda" (Grant Proposal Period: January 1, 2020 – December 31, 2022), based on the agreement fully executed on December 19, 2019. The END Fund provided technical input during study design and contributed to the review of study findings and the manuscript but was not involved in data analysis

**Competing interests:** The authors have declared that no competing interests exist.

## Findings

The overall prevalence of STH was 38.7% (95% CI: 37.9–39.4), highest among adults (46.1%, 95% CI: 44.8–47.3) and lowest among preschool-aged children (30.2%, 95% CI: 29.0–31.5). Species-specific prevalence was 27.0% for *Ascaris lumbricoides* (95% CI: 26.3–27.6), 11.6% for *Trichuris trichiura* (95% CI: 11.2–12.1), and 10.7% for hookworm (95% CI: 10.3–11.2). Moderate-to-heavy intensity (MHI) infections were detected in 8.1% of *Ascaris lumbricoides* (95% CI: 7.7–8.5), 0.8% of *Trichuris trichiura* (95% CI: 0.6–0.9), and 0.1% of hookworm (95% CI: 0–0.2). SCH prevalence was 1.7% (95% CI: 1.5–1.9) by KK and 27.2% (95% CI: 26.5–27.9) when trace results on POC-CCA were considered positive. Heavy *Schistosoma mansoni* infections were rare (0.1%, 95% CI: 0–0.1). Mixed-effects logistic regression (p < 0.05) showed that for STH, higher odds were associated with being single (AOR: 1.74), no education (AOR: 1.56), use of human excreta as manure (AOR: 1.43), unimproved water sources (AOR: 1.17), and proximity to marshlands (AOR: 1.17). Lower odds were seen among those with higher education (AOR: 0.55), unemployed (AOR: 0.34), self-employed or retired (AOR: 0.53), students (AOR: 0.54), those with deep toilets (AOR: 0.78), and those treating water consistently (AOR: 0.79). For SCH, higher odds were linked to being single (AOR: 1.61), no education (AOR: 1.41), proximity to lakes (AOR: 1.76) or rice fields (AOR: 1.31), use of treated (AOR: 1.32) or untreated (AOR: 1.60) excreta as manure, and living over an hour from a water source (AOR: 1.42).

## Conclusion

STH and SCH remain significant public health challenges in Rwanda, with certain regions and population groups still exceeding the elimination threshold as public health problems. Expanding MDA programs to include adults, improving sanitation and hygiene, ensuring universal access to clean water, and promoting community education on safe practices are essential for achieving sustainable control and elimination of these infections.

### Author summary

Worm infections, like soil-transmitted helminthiasis (STH) and schistosomiasis (SCH), continue to affect people of all ages in Rwanda. Despite efforts to control these infections through regular treatment programs, more up-to-date information on how common they are and what increases the risk of getting them is needed to improve control measures. This study examined the current situation across people aged 1 year and above in Rwanda.

We carried out a nationwide survey in all 30 districts of Rwanda, testing 17,360 people for worm infections. The results showed that 38.7% of people had

intestinal worms, with adults being the most affected (46.1%) and preschool-aged children the least affected (30.2%). The percentage of infected people varied widely between districts, ranging from 8.6% to 85.1%. Among the worms, roundworms were the most common (27.0%), followed by whipworms (11.6%) and hookworms (10.7%). Most infections were mild, with fewer cases of moderate-to-heavy infections, which are of greater concern due to their severe health consequences.

For schistosomiasis, the percentage of people infected ranged from 0% to 38.3% using the standard testing method and up to 74.6% when results from a newer method were included. Heavy intensity infections were rare, with only a small percentage (0.1%) of cases showing higher levels of infection.

We identified certain groups that were more likely to have these infections. For intestinal worms, a higher likelihood was seen among single people, those without education, those using treated human excreta as manure, relying on unsafe water sources, and living near marshy areas. On the other hand, people with higher education, clean water, better sanitation, and certain job types were less likely to have infections. For schistosomiasis, the likelihood of infection was higher for those who were single, without education, lived near lakes or rice farms, used human excreta as manure, and lived far from water sources.

Intestinal worms and schistosomiasis remain significant public health challenges in Rwanda. This study highlights the need for continued treatment programs, improved water, sanitation, and hygiene services, and targeted interventions to protect high-risk groups and areas. These measures are essential for reducing the burden of these infections and working toward their elimination.

## Introduction

The persistent burden of soil-transmitted helminthiases (STH) and schistosomiasis (SCH) in Rwanda, as in many sub-Saharan African countries, highlights the need for updated data to guide targeted strategies. STH, caused by *Ascaris lumbricoides* (roundworm), *Ancylostoma duodenale*/*Necator americanus* (hookworm), and *Trichuris trichiura* (whipworm), affects over 1.5 billion people, or 24% of the global population, while SCH impacts over 251 million individuals globally who require preventive treatment, with 90% of cases in Africa [1,2].

STH are transmitted through ingestion of eggs (in the case of *Ascaris* and *Trichuris*) or skin penetration by larvae (in the case of hookworms), often due to poor sanitation and hygiene. SCH, caused by *Schistosoma* species, is acquired through skin contact with freshwater containing cercariae released by infected snails, particularly in areas near lakes, rivers, and wetlands [3]. Both infections contribute to a substantial public health burden, particularly among school-aged children (SAC) and agricultural communities. They are associated with anemia, malnutrition, stunting, cognitive impairment, and, in chronic SCH cases, hepatosplenic or urogenital complications [4,5].

In Rwanda, a national survey in 2007–2008 reported an STH prevalence of 65.8% and a SCH prevalence of 2.7%. In response, the Rwandan Neglected Tropical Diseases (NTD) Programme launched mass drug administration (MDA) in 2008, targeting preschool-aged children (pre-SAC), SAC, and selected adults. Albendazole (ALB) or mebendazole (MBZ) was administered biannually, while praziquantel (PZQ) was co-administered in districts with SCH prevalence over 10%. These programs achieved high coverage, with over 90% for ALB (or MBZ) and 75% for PZQ [6–8].

By 2014, an impact assessment demonstrated significant progress. STH prevalence among SAC dropped from 65.8% to 45.2%, though district-level variations remained high, ranging from 2.1% to 89.6%. *Ascariasis* was the most prevalent STH infection (37.2%), followed by *Trichuriasis* (22.8%), while hookworm prevalence was 4.5% [9]. For SCH, *Schistosoma*

*mansoni* prevalence was 1.9% using the Kato-Katz (KK) method and 7.4% using the Point-of-Care Circulating Cathodic Antigen (POC-CCA) test—a rapid urine-based assay that detects schistosome antigens [10–12]. Trace results were classified as negative in that analysis. A total of 127 sectors—administrative units below the district level—were identified as endemic for SCH. Following the 2014 remapping, annual community- and school-based MDA for STH continued nationwide, targeting pre-SAC, SAC, pregnant women, and breastfeeding mothers. In addition, PZQ was administered annually in the 127 SCH-endemic sectors [7].

Since the 2014 survey, Rwanda has undergone significant environmental and social changes alongside continuous high-coverage MDA, highlighting the need for a reassessment of STH and SCH. Access to improved water sources increased from 73.8% in 2014 to 80.4% in 2020, while the percentage of households with improved sanitation facilities saw a modest increase from 71.2% to 72.2% [13,14]. At the same time, developments such as expanded agricultural water use, hydroelectric projects, and the creation of artificial lakes may have inadvertently fostered conditions that support SCH transmission.

Evidence suggests that STH infections are more prevalent in adults, particularly in areas where school-based MDA programs have reduced infections in children but sustained transmission [15,16]. Additionally, assessing SCH in adults is crucial for informing treatment strategies [17]. However, MDA programs and evaluations in Rwanda have primarily focused on pre-SAC and SAC, with limited data on adults [8,9,18–20]. Moreover, while the national health management information system captures STH and SCH cases diagnosed at facilities, it excludes individuals with low-intensity or asymptomatic infections who are less likely to seek care, underestimating transmission [21]. Additionally, previous studies that included adults were not nationally representative and often excluded SCH [22]. To address these gaps, comprehensive, updated, and representative data were needed to inform progress toward the World Health Organization's (WHO) elimination targets of <2% prevalence of moderate-to-heavy intensity STH infections and <1% heavy-intensity SCH infections [23]. In response, this study reassessed the prevalence and risk factors of STH and SCH across the population aged 1 year and above in Rwanda to optimize control strategies, target interventions, and guide efforts toward eliminating these infections as public health problems.

## Methods

### Ethics statement

This study was conducted following ethical guidelines and received approval from the Rwanda National Ethics Committee (Approval No: 652/RNEC/2020). A written informed consent was obtained from all participants before data collection began. For participants under 18 years of age, written informed consent was obtained from parents or guardians, and assent was obtained from children aged 12–17 years. Participation was voluntary, and participants could withdraw at any time without consequences. For schoolchildren, permissions were sought from parents and schools to allow them to participate briefly, with transportation arranged to ensure their timely return to school.

Following the interview, those aged two years and above received a single dose of Albendazole (400 mg), while younger children received Mebendazole (500 mg). Samples were tested, and participants were informed promptly once the result was available. If participants needed to leave early, they could consent to have their results delivered via community health workers. Participants who tested positive for SCH and were aged five years or older received a single 600 mg dose of praziquantel while those under five years were referred to health centers for treatment per the national guidelines.

### Study design

This study conducted a nationwide, community-based cross-sectional survey using an adapted precision mapping approach to assess the prevalence and identify risk factors associated with SCH and STH. Precision mapping focuses

on sampling at a finer geographic scale, targeting more schools or villages within smaller administrative units to better capture the variability in SCH prevalence [24]. Following recommended guidelines, SCH and STH were mapped together, ensuring a comprehensive assessment [23,25–27].

## Study setting

Rwanda, a low-income, landlocked African nation, spans approximately 26.3 square kilometers and with a population of around 12.6 million in 2020, estimated to reach 13.8 million in 2024, resulting in one of the highest population densities in Africa at 501 residents per square kilometer [28,29]. The majority of the population, 72.1%, live in rural areas, with 69% engaged in agriculture. Among households, 63% practice crop farming, and 50% own at least one type of livestock [28]. In 2020, the GDP per capita was estimated at $816 and increased to $1,040 in 2023, with agriculture contributing 27% [30,31].

Rwanda is divided into five provinces: North, East, West, South, and the City of Kigali. Each province is further sub-divided into districts, with a total of 30 districts nationwide. These districts are further divided into sectors (416 sectors in total), which are responsible for implementing development programs, delivering services, and promoting governance and social welfare. Sectors are subdivided into cells (2,148 cells), and each cell comprises villages, totaling 14,815 villages [32]. The Rwandan health system follows a traditional hierarchical cascade, structured as community, health center, district hospital, and referral and specialized hospitals [6,7].

## Participants

The study participants included pre-SAC (1–4 years old), SAC (5–15 years old), and adults (16 years and above). Children aged from one year old were included because they are targeted in deworming programs, and surveys in similar settings to Rwanda have found that these children are also at risk of SCH infection when their parents or caregivers take them to high-risk areas [33].

## Variables

**Dependent variables.** *Schistosomiasis status and intensity: S. mansoni* was assessed by examining stool samples for schistosome eggs using the KK technique (duplicate slides), following WHO guidelines [34,35], and by testing urine samples using the POC-CCA method. For the main analysis, individuals were considered SCH-positive if they tested positive by either KK (presence of at least one *S. mansoni* egg) or POC-CCA, with trace results included as positive. This approach was selected for programmatic reasons, as the program aims to eliminate SCH by targeting all potentially infected individuals, including those with low-level or asymptomatic infections.

Additionally, results for KK alone were presented with *S. mansoni* intensity levels, classified by eggs per gram (epg) of feces. Light-intensity was defined as 1–99 epg, moderate-intensity as 100–399 epg, and heavy-intensity as ≥400 epg, as per WHO guidelines [34,35]. Furthermore, to illustrate variations in SCH prevalence based on test interpretation, we also presented SCH status when trace results for POC-CCA were considered negative and SCH status when an individual tested positive by either KK or POC-CCA but with trace results considered negative.

*Schistosoma haematobium* infection was assessed through gross haematuria and dipstick testing. In cases with hae-maturia, urine samples were centrifuged and examined microscopically for *S. haematobium* ova. As no ova were detected and haematuria was not a primary focus of this study, infection intensity classification for *S. haematobium* was not applicable and was therefore not included.

*Any STH Status and Intensity:* Any STH status indicates the presence of any STH infection, including *Ascaris lumbricoides*, hookworm, and *Trichuris trichiura*. Participants were classified as having an STH infection if their stool samples tested positive for at least one egg of any helminth species using the KK technique with duplicate slides.

The intensity of STH infections was classified based on the number of epg of feces per WHO guideline [34,35]. For *Ascaris lumbricoides*, light-intensity infections were defined as 1–4,999 epg, moderate-intensity as 5,000–49,999 epg, and heavy-intensity as ≥50,000 epg. For *Trichuris trichiura*, light-intensity was 1–999 epg, moderate-intensity was 1,000–9,999 epg, and heavy-intensity was ≥ 10,000 epg. For hookworms, light-intensity was 1–1,999 epg, moderate-intensity was 2,000–3,999 epg, and heavy-intensity was ≥ 4,000 epg [34].

### Independent variables

Age was recorded as the participant's actual age and categorized into three groups: pre-SAC (1–4 years), SAC (5–15 years), and Adults (16 years and above). Certain variables were collected only for participants aged 10 years and above, including marital status (single, married, divorced, or widowed), education level (no education, primary, secondary, tertiary), and main occupation, defined as the participant's primary source of income or work.

Other independent variables included the main source of household water, classified as improved (e.g., piped water, wells, and rainwater) or non-improved (all other sources). Additionally, the presence of a toilet facility was recorded, along with its depth if present (<3 meters deep or ≥3 meters deep). We also documented whether households used human feces as manure and whether they treated water before drinking, as well as the primary method of treatment among those who reported always treating drinking water. Participants' knowledge about SCH and STH, including transmission, symptoms, and prevention, was also evaluated.

### Data sources and measurement

**Data collection tools.** The field data collection took place from 13th November to 24th December 2020, five months later after the MDA. The survey used a questionnaire to collect information on risk factors, knowledge, and attitudes related to STH and SCH among individuals aged 10 years and above, along with household conditions. It covered demographic details such as age, sex, education, occupation, and household attributes, as well as water sources, sanitation practices, and the use of human waste as manure, focusing on factors that increase infection risk. Additional questions explored exposure to water bodies, potential SCH transmission sources, and hygiene behaviors like handwashing and water treatment. Participants were also asked about infection transmission, symptoms, and their attitudes toward STH and SCH.

The questionnaire was initially developed in English and translated into Kinyarwanda, the local language, by bilingual team members. It was reviewed by other fluent researchers to ensure linguistic accuracy and cultural relevance, then digitized using an Open Data Kit (ODK)-based system for administration on smartphones. Pre-testing was conducted in two districts to validate its content and ensure consistency in data collection.

### Recruitment and data collection procedures

Data collection began with village leaders and community health workers contacting participants from randomly selected households. These local representatives coordinated with health centers to schedule convenient dates and locations for data collection. Upon arrival, participants provided consent and were assigned a card with a unique code. They were then given containers for stool and urine sample collection.

When participants returned with their samples, laboratory technicians verified the samples to ensure that the identifiers matched the participant codes and logged the information into a digital system. If participants were unable to provide a sample immediately, they were encouraged to return it later.

After submitting their samples, participants took part in an interview session. Trained data collectors, who were also nurses, conducted structured interviews in Kinyarwanda using a digital questionnaire. All data were uploaded daily to a central server hosted by the Rwanda Biomedical Centre, following verification by the team leader for questionnaire data and by senior laboratory technicians for laboratory results. All procedures adhered to local data protection regulations.

## Laboratory testing and quality assurance

Laboratory testing for STH and SCH followed standardized procedures. For STH, including *A. lumbricoides*, *T. trichiura*, and hookworm, stool samples from each participant were analyzed using duplicate Kato-Katz slides. These slides were examined under a light microscope, and egg counts were recorded to determine infection intensity. Due to the fragile nature of hookworm eggs, which can disintegrate after 40–60 minutes, the slides were initially read within 20 minutes to capture hookworm eggs, with a second reading later for other parasites [36].

For SCH, stool samples were analyzed using the KK technique, while urine samples were tested using the POC-CCA test and dipsticks for haematuria. The POC-CCA is a rapid, urine-based assay that detects *Schistosoma mansoni* antigens and is well suited for field use due to its ease of administration and high sensitivity, particularly for low-intensity infections. One drop of urine was applied to the test cassette, followed by a drop of buffer solution; results were read after 20 minutes and scored as negative, trace, or positive (1+ to 3+), based on band intensity. In cases where haematuria was detected by dipstick, additional urine testing was conducted using centrifugation and microscopic examination to assess for *S. haematobium* infection antigens [10–12].

To ensure accuracy and quality in laboratory testing, for microscopy, a refresher training was conducted by the senior technician form the National Reference Laboratory (NRL) for qualified laboratory technicians experienced in KK techniques and microscopy from health facilities. Positive slides with SCH and STH species and bench aids with image of parasites 'eggs were used during the training. During field activities, two laboratory technicians read duplicate slides (one read A another read B). A qualified quality controller (a senior technician from NRL or referral hospital) in each team conducted quality control of 20% randomly selected slides (+/- 12 slides). Any discrepancies exceeding 10% (in terms of the type or number of eggs) between the senior technician's readings and those of the field microscopists were required to be reported to the survey coordinator; however, no discrepancies >10% identified.

For POC-CCA tests, intra-reader reliability checks were implemented. After the initial reading, a quality controller and two additional technicians re-read 10% of the CCA cassettes results on daily basis. Results were compared, and discrepancies over 10% but less than 20% were discussed to reach consensus. Larger discrepancies (over 20%) triggered external review and refresher training.

## Bias and mitigation measures

To minimize potential biases, the study implemented several strategies. Ten data collection teams, each consisting of lab technicians, nurses, and quality controllers, were supported by local health staff, with field team leaders ensuring standardized procedures across all locations. A five-day training workshop and pre-test in two districts refined logistics, validated tools, and ensured consistency in data collection. Selection bias was addressed by recruiting participants from randomly selected households, coordinated by community leaders to ensure diversity. Reporting bias was minimized through structured interviews conducted by trained nurses using a clear questionnaire, reducing ambiguity and recall errors.

Sample handling bias was prevented by assigning unique codes to participants for accurate tracking of stool and urine samples, with staff verifying identifiers. Standardized procedures, including duplicate KK slides and specific timing for hookworm readings, helped reduce measurement bias. Quality assurance measures, such as daily checks by team leaders and senior laboratory technicians and intra-reader reliability checks, ensured consistent testing. Data was recorded by aids of an Open Data Kit (ODK)-based system, with daily uploads to a central server for regular quality checks, review, and cleaning, maintaining data accuracy and completeness.

## Study size

The sample size determination followed the WHO protocol for mapping neglected tropical diseases amenable to preventive chemotherapy in the African region, which recommends sampling 50 participants per village and at least five villages in each

ecological zone [37]. In this study, villages the smallest administrative units in Rwanda, each with about 150–200 households served as the survey areas. The survey covered all 30 districts, with five sectors selected per district, resulting in 150 sectors. Within each sector, two villages were chosen, resulting in a total of 10 villages per district, totaling 300 villages. In each village, 60 participants were selected, aiming for 600 participants per district and a nationwide total of 18,000 participants.

## Sampling process

A three-stage sampling design was used to select study villages, focusing on areas likely to support schistosomiasis transmission due to their proximity to water bodies or wetlands.

   **Stage 1: Determination of eligible villages:** In the first stage, villages adjacent to water bodies or wetlands were identified, given the focal transmission patterns of schistosomiasis, which typically occur in such environments. The sampling process was informed by both known and suspected transmission areas, utilizing available epidemiological data as stipulated by the WHO, which recommends purposive sampling considering sub-districts instead of random sampling due to the high focality of schistosomiasis transmission [37]. Geographic Information System (GIS) software and data from the Rwanda Environment Management Authority (REMA) and the Ministry of Environment/Water Department were used to identify 6,165 villages located near significant water bodies or wetlands. Villages were excluded if they were near very small wetlands (less than 1 hectare) or isolated wetlands, assuming that these do not retain sufficient water throughout the year to support transmission. Villages located in wetlands permanently converted to agricultural use, such as tea plantations or other cash crops were also excluded. This determination was made using data from the Ministry of Agriculture and Animal Resources (MINAGRI), the National Agricultural Export Development Board (NAEB), and cross-referenced with satellite imagery from Google Earth [38,39]. Additionally, villages within protected areas, such as national parks, natural forests, and RAMSAR sites, were excluded from the sampling.

   **Stage 2: Selection of villages:** During the second stage, villages at high risk of schistosomiasis transmission were identified based on ecological and epidemiological factors. Data were collected from all eligible villages through structured phone interviews with local officials, such as sector agriculture officers and village leaders, who provided extensive local knowledge. The data collection utilized an adapted WHO form designed to improve the use of available prevalence data and support subdistrict-level planning for schistosomiasis control [40]. The collected data were used to rank the villages by risk level, with the criteria presented in Table 1.

**Table 1. Grading criteria for villages to determine their risk for schistosomes.**

| Factor | Condition | Grade |
|---|---|---|
| Type of water body | Villages adjacent to lakes, water dams, ponds, fishponds, marshes, or wetlands | 2 |
| | Villages near rivers or water streams | 1 |
| | Villages near torrents or surfaces with non-permanent water | 0 |
| Type of human activity | Swimming, fishing, rice cultivation, water fetching, irrigation, livestock watering | 2 |
| | Cultivation of other crops or vegetation, other activities | 1 |
| | No water-related activities | 0 |
| Estimated number of population exposed | More than 100 individuals exposed | 2 |
| | 1-100 individuals exposed | 1 |
| | No individuals exposed | 0 |
| Proximity of households to water bodies | Households located less than 500 meters from a water body | 2 |
| | Households located more than 500 meters from a water body | 1 |
| | No adjacent water body | 0 |
| Confirmed cases of schistosomiasis | Confirmed cases by survey or routine reporting | 1 |
| | No confirmed cases | 0 |

Villages that scored high were considered high risk. A total of 3,100 villages were ranked as high risk according to these factors. The high-risk villages were grouped by district and sector. Within each district, five sectors were selected. From each selected sector, two villages were chosen from the pool of high-risk villages, resulting in a total of 10 villages per district and 300 villages nationwide (Fig 1). The selection process was conducted visually, considering the spatial distribution and ecosystem characteristics of the high-risk villages to ensure that the selected sectors and villages were well-distributed across each district, capturing the ecological and geographical variations within the districts.

**Stage 3: Selection of households and study participants:** For each selected village, health centers conducted a census of all households, recording the name, sex, and age of every individual. Households with members representing all three age groups (1 – 4 years, 5–15 years, and 16 + years) were included in the sampling frame. Systematic sampling was used to select 20 households per village to ensure 60 participants per village. The sampling interval was calculated by dividing the total number of households by the number to be selected. The first household was chosen randomly within this interval, and subsequent households were selected systematically, resulting in a total of 6,000 households across all villages.

Within each of the 20 selected households per village, participants were chosen using simple random sampling to ensure representation across the three age groups. In the first household, one individual from each age group was randomly selected. If there were multiple individuals in an age group, names were written on pieces of paper, mixed, and one name was drawn at random. For subsequent households, participants were selected with consideration to balance the number of males and females per age group in each village [34]. Individuals with mental disabilities preventing them from responding to the survey questions were excluded from the study.

**Consideration for STH:** It was assumed that STH transmission is relatively homogeneous within districts or sub-districts, based on previous national mapping efforts that did not reveal clear or consistent variation at finer geographic scales [8,9,20,41]. Consequently, the villages and participants selected for SCH were also considered suitable for

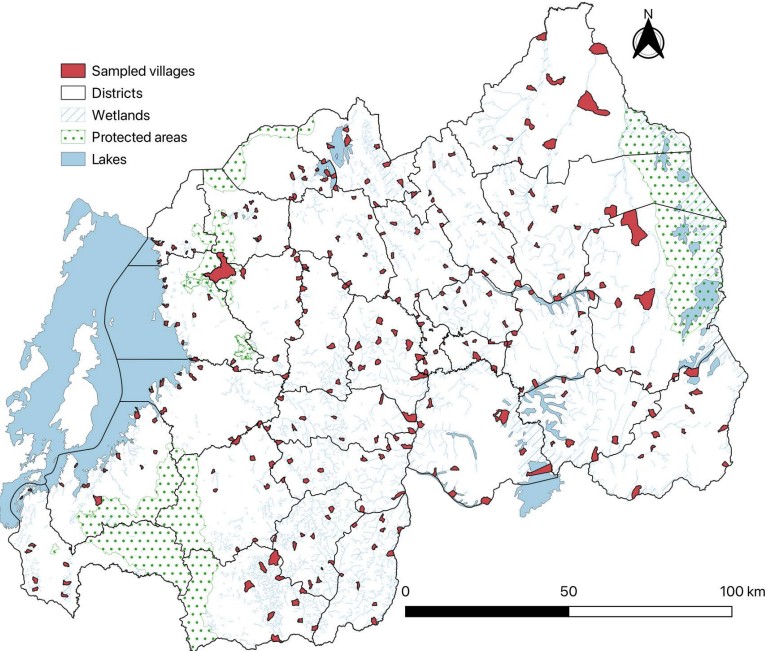

**Fig 1. Selected villages across districts highlighted in red.** The base layer consists of shapefiles provided by the National Institute of Statistics of Rwanda and publicly available through the Africa GeoPortal, powered by Esri: https://rwanda.africageoportal.com/search?q=rwanda.

assessing STH. This approach was deemed appropriate given the large sample size, broad geographic coverage of the selected villages, and the need for operational efficiency. Therefore, no additional sampling was conducted specifically for STH.

## Statistical methods

Data were downloaded in Excel format and subsequently exported to R for cleaning and analysis. All data cleaning and analysis were conducted using R version 4.4.1. The dataset was stored on a password-protected computer, ensuring secure access. Only the research team had access to the data. All identifiable information was removed to maintain confidentiality and protect participant privacy.

Descriptive statistics were used to summarize the data. Frequencies and percentages were calculated to describe the study population and the prevalence of SCH and STH infections overall and by species. Infection intensity for SCH and STH was categorized as light, moderate, or heavy according to WHO guidelines, based on EPG values [34,35]. Additionally, a sensitivity analysis was conducted to adjust prevalence estimates for age; the methods and results of the age-adjusted analysis are presented in S1 Text.

To explore associations between independent variables and outcomes, chi-square tests were conducted to identify variables significantly associated with each outcome at a significance level of $p < 0.05$. Variables with p-values <0.05 in bivariate analyses and additional covariates identified from the literature were included in the multivariable mixed-effects logistic regression models with districts as random intercepts to determine risk factors for STH and SCH in separate models. The STH model was adjusted for age category, marital status, education level, main occupation, toilet facility type, soap and water availability, main water source, water treatment behavior, use of human excreta as fertilizer, and proximity to marshlands (for rice and other uses). The SCH model was adjusted for age category, marital status, education level, main occupation, main water source, time to water source, use of human excreta as fertilizer, proximity to various water bodies (lake, river, pond/dam, marshlands, other), and walk time to the nearest water body. To avoid multicollinearity, correlations between variables were tested beforehand, ensuring only non-correlated variables were included. The significance level for the models was set at $p < 0.05$. Adjusted odds ratios (ORs), 95% confidence intervals (CIs), and p-values were reported for each variable included in the final models.

***Comparison of STH and S. mansoni prevalence, 2008–2020.*** To assess changes in prevalence across surveys, we focused on SAC, as previous surveys were conducted in this age group: 10–16 years in schools in 2008, 9–18 years in schools in 2014, and 5–15 years in the community in 2020. Historical prevalence data and the number of individuals tested for each parasite species in 2008 and 2014 were obtained from national program survey reports and related publications [9,41]. For each infection, absolute differences in prevalence between survey years (2008–2014 and 2014–2020) were calculated, along with 95% confidence intervals (CIs). CIs were estimated using the standard error of the difference between two survey proportions, based on the standard normal approximation (Wald method), which assumes independent samples and sufficiently large sample sizes. For S. *mansoni*, results from KK and POC-CCA tests were presented separately. For the POC-CCA, trace results were considered positive in both the 2014 and 2020 datasets; the 2008 survey did not include the POC-CCA test. No *S. haematobium* infections were detected in any of the survey years. All results are reported as percentage point differences. Details of the estimates from previous surveys for each STH and SCH are available in the S4 Table.

**Knowledge and attitudes assessment.** A structured knowledge and attitudes module was included to assess community awareness and perceptions of SCH and STH among participants aged 10 years and above, who were able to respond to knowledge-related questions. Participants were first asked if they had heard of either condition. Those who responded "yes" were asked about sources of information (e.g., churches, community health workers, health facilities, media, schools, parents/elders, or other), time since last receiving information, knowledge of transmission, and awareness of treatment.

Knowledge of transmission was defined as correctly identifying key pathways contact with contaminated water for SCH, and poor hygiene, open defecation, or unsafe food or water for STH. Treatment awareness was based on identifying albendazole, mebendazole, or similar deworming medications.

Attitudes were assessed using Likert-scale questions. Responses of "agree" or "strongly agree" were coded as "Agree," while all others were classified as "Disagree." For SCH, attitude items covered perceived severity, importance of screening, treatment uptake, and care-seeking for symptoms. For STH, they included beliefs about preventability, use of untreated human excreta as fertilizer, and traditional medicine.

Responses were stratified by age group school-aged children (SAC: 10–15 years) and adults (≥16 years) and summarized as frequencies and percentages. "No" responses and missing data were excluded from the denominator for each indicator.

## Results

### Sociodemographic characteristics

The study included 17,765 participants across five provinces: East (n = 4,115; 23.2%), Kigali City (n = 1,768; 10.0%), North (n = 2,981; 16.8%), South (n = 4,750; 26.7%), and West (n = 4,151; 23.4%). The sample comprised 9,582 (53.9%) females and 8,183 (46.1%) males. Participants were categorized as preschool-aged children (1–4 years; n = 5,573, 31.4%), school-aged children (5–15 years; n = 6,166, 34.7%), and adults (≥16 years; n = 6,026, 33.9%).

Of the 8,559 participants reporting marital status, 56.7% were married, 36.8% were single, and 6.5% were divorced, separated, or widowed. Among 8,563 respondents, 54% had no formal education, 41.9% had primary education, and 4.1% had secondary or higher education. Most (58.4%) of 8,564 participants who reported their main occupation were engaged in agriculture or farming, while 31.3% were students.

At the household level, 87.4% of the 4,906 surveyed households had a toilet facility with a depth of at least 3 meters, and 57.8% of the 4,841 respondents reported always having soap and water available. In terms of water treatment practices, 45% of respondents indicated they never treated water before drinking, with boiling being the most common method (92.9%) among those who did treat their water. Regarding the use of human excreta as manure, 73.1% of households did not use these, while 22% reported treating them before use.

About proximity to water bodies, 53.3% of households were located near a river, and 64.3% were near marshlands used for activities other than rice farming. Detailed sociodemographic characteristics can be found in Table 2.

### The prevalence and intensity of STH and *S. mansoni*

Table 3 shows the variation in the prevalence of *STH* and *S. mansoni* across different age groups. The overall prevalence of any STH infection was 38.7% (95% CI: 37.9–39.4), with the highest prevalence observed among adults (46.1%, 95% CI: 44.8–47.3) and the lowest among pre-SAC (30.2%, 95% CI: 29.0–31.5). The prevalence of *A. lumbricoides*, *T. trichiura*, and hookworm was 27.0%, 11.6%, and 10.7%, respectively. SAC exhibited the highest prevalence of *A. lumbricoides* (30.5%, 95% CI: 29.4–31.7) and *T. trichiura* (15.0%, 95% CI: 14.1–15.9), while adults had the highest estimated prevalence of hookworm (21.3%, 95% CI: 20.2–22.3).

The overall prevalence of *S. mansoni* using the KK method was 1.7% (95% CI: 1.5–1.9). When including positives identified by either KK or POC-CCA (with trace results considered positive), the prevalence increased to 27.2% (95% CI: 26.5–27.9), with the highest prevalence observed among pre-SAC (35.4%, 95% CI: 34.1–36.7). No *Schistosoma haematobium* was detected in any tests. Adjusted prevalence estimates for STH and *S. mansoni* are provided in S1 Text.

Table 4 presents the intensity of infections with *STH* and *S. mansoni*. Heavy infections were rare, with *A. lumbricoides* highest at 0.5% (95% CI: 0.4–0.7), followed by *Hookworm at 0.1% (95%CI: 0 - 0.1])* and *T. trichiura* at 0.02% (95% CI: 0.0–0.1), and *S. mansoni* at 0.1% (95% CI: 0.0–0.1). Moderate infections were highest for *A. lumbricoides* at 7.5% (95% CI: 7.1–7.9), while light infections dominated, with *A. lumbricoides* at 18.9% (95% CI: 18.3–19.5), *T. trichiura* at 10.9%

**Table 2. Individuals' and households' sociodemographic characteristics.**

| Variables | Categories | n | % |
|---|---|---|---|
| *Individual social demographics* | | | |
| Province (N = 17765) | East | 4115 | 23.2 |
| | Kigali City | 1768 | 10 |
| | North | 2981 | 16.8 |
| | South | 4750 | 26.7 |
| | West | 4151 | 23.4 |
| Sex of the participant (N = 17765) | Female | 9582 | 53.9 |
| | Male | 8183 | 46.1 |
| Age categories (N = 17765) | pre-SAC (1–4) | 5573 | 31.4 |
| | SAC (5–15) | 6166 | 34.7 |
| | Adults (16 and above) | 6026 | 33.9 |
| Marital status (N = 8559) | Single | 3151 | 36.8 |
| | Married | 4850 | 56.7 |
| | Divorced/Separated/Widowed | 558 | 6.5 |
| Education level (N = 8563) | No education | 4624 | 54 |
| | Primary | 3591 | 41.9 |
| | Vocational or literacy training | 15 | 0.2 |
| | Secondary and higher | 333 | 3.9 |
| Main occupation (N = 8564) | Agriculture/Farmer | 5002 | 58.4 |
| | Employed (formal/informal) | 399 | 4.7 |
| | Self-employed/Retired | 135 | 1.6 |
| | Student | 2682 | 31.3 |
| | Unemployed/Other | 346 | 4 |
| *Household-level variables* | | | |
| Has toilet facility (N = 4906) | No | 65 | 1.3 |
| | Yes, < 3 meters deep | 555 | 11.3 |
| | Yes, 3 + meters deep | 4286 | 87.4 |
| Always have soap and water (N = 4841) * | No | 2045 | 42.2 |
| | Yes | 2796 | 57.8 |
| Main water source (N = 4906) | Improved source | 1933 | 39.4 |
| | Non-improved source | 2973 | 60.6 |
| Treat water before drinking (N = 4906) | Never | 2210 | 45 |
| | Yes, sometimes | 1108 | 22.6 |
| | Yes, always | 1588 | 32.4 |
| Main water treatment methods *(among those who always treat water)* (N = 1588) | Boiling | 1475 | 92.9 |
| | Chemical | 34 | 2.1 |
| | Filtering | 62 | 3.9 |
| | Others | 17 | 1.1 |
| Time to closest water source (round trip) (N = 3143) * | On premises or <30 min | 1690 | 53.8 |
| | 30 - 60 min | 884 | 28.1 |
| | More than 1 hour | 569 | 18.1 |
| Use human excreta as fertilizer (N = 4841) * | No | 3539 | 73.1 |
| | Yes, and treat the fertilizer used | 1066 | 22 |
| | Yes, does not treat fertilizer used | 236 | 4.9 |

*(Continued)*

**Table 2.** (Continued)

| Variables | Categories | n | % |
|---|---|---|---|
| Nearest water bodies (N = 4906) *(Participants could have more than one type.)* | Lake | 1202 | 24.5 |
| | River | 2616 | 53.3 |
| | Pond/Dam | 1165 | 23.7 |
| | Marshlands for rice farming | 1070 | 21.8 |
| | Marshlands for other activities | 3153 | 64.3 |
| | Other water bodies | 222 | 4.5 |
| Walk time to closest water bodies (N = 4906) | 0-20 min | 2871 | 58.5 |
| | 21-40 min | 1387 | 28.3 |
| | 41 + min | 648 | 13.2 |

*There were missing values for some participants.

**Table 3. Prevalence of STH and *S. mansoni*.**

| Species | Overall | | | pre-SAC (1–4 years) | | | SAC (5–15 years) | | | Adults (16 and above) | | |
|---|---|---|---|---|---|---|---|---|---|---|---|---|
| | % | 95% CI | Tested | % | 95% CI | Tested | % | 95% CI | Tested | % | 95% CI | Tested |
| **Soil-Transmitted Helminths** | | | | | | | | | | | | |
| Any STH | 38.7 | [37.9 - 39.4] | 17360 | 30.2 | [29 - 31.5] | 5309 | 38.8 | [37.6 - 40.1] | 6104 | 46.1 | [44.8 - 47.3] | 5947 |
| *A. lumbricoides* | 27 | [26.3 - 27.6] | 17348 | 24.5 | [23.3 - 25.6] | 5307 | 30.5 | [29.4 - 31.7] | 6098 | 25.6 | [24.5 - 26.7] | 5943 |
| Hookworm | 10.7 | [10.3 - 11.2] | 17338 | 4.3 | [3.8 - 4.9] | 5303 | 6.1 | [5.5 - 6.7] | 6096 | 21.3 | [20.2 - 22.3] | 5939 |
| *T. trichiura* | 11.6 | [11.2 - 12.1] | 17352 | 8.4 | [7.7 - 9.2] | 5308 | 15.0 | [14.1 - 15.9] | 6101 | 11.0 | [10.3 - 11.9] | 5943 |
| ***Schistosoma mansoni*** | | | | | | | | | | | | |
| *S. mansoni* (KK) | 1.7 | [1.5 - 1.9] | 17342 | 0.6 | [0.4 - 0.9] | 5304 | 2.4 | [2 - 2.8] | 6100 | 1.8 | [1.5 - 2.2] | 5938 |
| *S. mansoni* (CCA - Trace positive) | 27.0 | [26.3 - 27.7] | 17645 | 35.6 | [34.4 - 36.9] | 5485 | 25.6 | [24.5 - 26.7] | 6151 | 20.5 | [19.5 - 21.6] | 6009 |
| *S. mansoni* (KK or **POC**-CCA - Trace positive) | 27.2 | [26.5 - 27.9] | 17719 | 35.4 | [34.1 - 36.7] | 5542 | 25.9 | [24.8 - 27] | 6159 | 20.9 | [19.9 - 22] | 6018 |
| *S. mansoni* (**POC**-CCA - Trace negative) | 10.9 | [10.5 - 11.4] | 17645 | 15.5 | [14.6 - 16.5] | 5485 | 10.1 | [9.3 - 10.9] | 6151 | 7.6 | [6.9 - 8.3] | 6009 |
| *S. mansoni* (KK or POC-CCA - Trace negative) | 11.4 | [11 - 11.9] | 17719 | 15.6 | [14.7 - 16.6] | 5542 | 10.7 | [9.9 - 11.5] | 6159 | 8.3 | [7.6 - 9] | 6018 |

(95% CI: 10.4–11.3), and *Hookworm* at 10.6% (95% CI: 10.1–11.1). Combined moderate and heavy infections were highest for *A. lumbricoides* at 8.1% (95% CI: 7.7–8.5).

Fig 2 presents the site-specific geographical distribution of prevalence for *A. lumbricoides*, Hookworm, and *T. trichiura* across Rwanda. For A. lumbricoides, higher prevalence clusters are observed in localized areas, particularly in the northern and southern regions. Hookworm prevalence is generally lower across the country, with most areas showing prevalence levels below 10% and only a few regions reaching 20–29.9%. T. *trichiura* exhibits a similar pattern to A. lumbricoides, with high prevalence areas (≥50%) concentrated in the northern regions, although many regions have prevalence levels below 10%. Additionally, regarding progress toward the elimination of STH as a public health problem, a detailed analysis showed that in 2020, fourteen districts had villages with an MHI prevalence of ≥2% for at least one STH species among SAC.

**Table 4. Intensity of STH and *S. mansoni*.**

| Species | Light | | | Moderate | | | Heavy | | | Moderate & Heavy | | | Mean* EPG | 95% CI mean EPG |
|---|---|---|---|---|---|---|---|---|---|---|---|---|---|---|
| | n | % | 95% CI | n | % | 95% CI | n | % | 95% CI | n | % | 95% CI | | |
| *A. lumbricoides* | 3283 | 18.9 | [18.3 - 19.5] | 1302 | 7.5 | [7.1 - 7.9] | 95 | 0.5 | [0.4 - 0.7] | 1397 | 8.1 | [7.7 - 8.5] | 6589.1 | [6203.1 - 6975.2] |
| Hookworm | 1836 | 10.6 | [10.1 - 11.1] | 13 | 0.1 | [0 - 0.1] | 11 | 0.1 | [0 - 0.1] | 24 | 0.1 | [0.1 - 0.2] | 210.4 | [167.1 - 253.8] |
| *T. trichiura* | 1885 | 10.9 | [10.4 - 11.3] | 129 | 0.7 | [0.6 - 0.9] | 4 | 0.02 | [0 - 0.1] | 133 | 0.8 | [0.6 - 0.9] | 324.1 | [283.8 - 364.5] |
| *S. mansoni* - KK | 225 | 1.3 | [1.1 - 1.5] | 54 | 0.3 | [0.2 - 0.4] | 9 | 0.1 | [0 - 0.1] | 63 | 0.4 | [0.3 - 0.5] | 92.2 | [74.3 - 110] |

*The mean number of eggs per gram (EPG) of feces was calculated among individuals who tested positive.

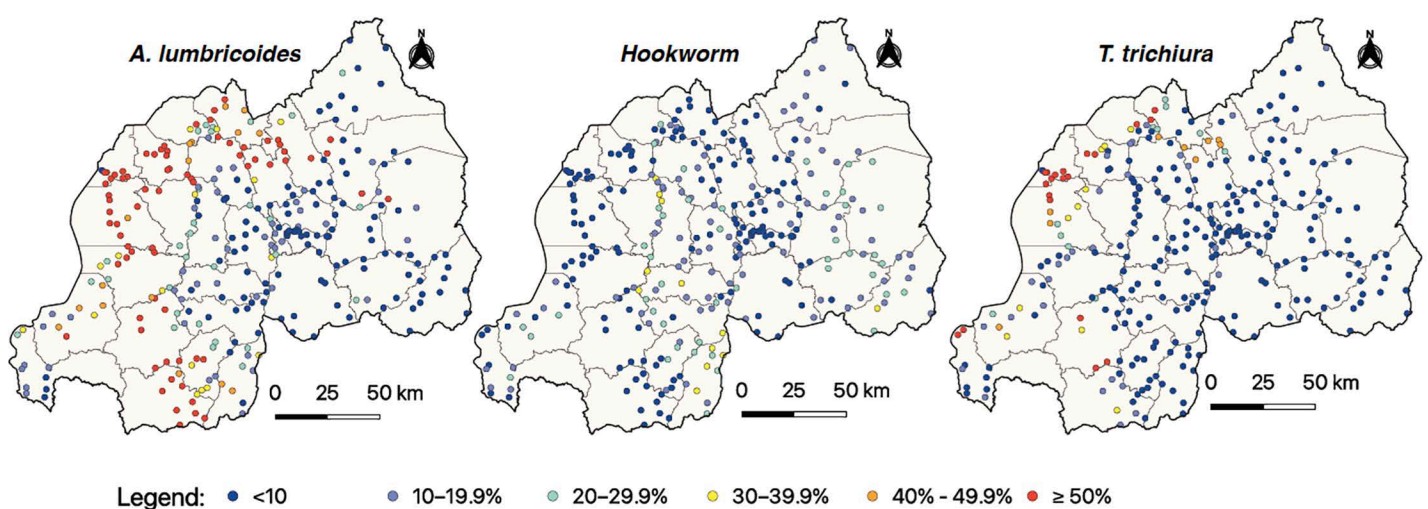

**Fig 2. Site-specific geographical distribution of STH prevalence across Rwanda.** The base layer consists of shapefiles provided by the National Institute of Statistics of Rwanda and publicly available through the Africa GeoPortal, powered by Esri: https://rwanda.africageoportal.com/search?q=rwanda.

Fig 3 shows the geographical distribution of S. *mansoni* prevalence. Using the KK method, most sites had low prevalence levels (<10%), with a few showing higher prevalence (10–49.9%) and none exceeding 50%. In contrast, the combined diagnostic approach (KK or POC-CCA) revealed significantly higher prevalence, with many sites reporting levels of 10–49.9% and several exceeding 50% across the country. Regarding progress toward the elimination of SCH as a public health problem, a detailed analysis showed that in 2020, six districts had test sites with ≥1% heavy infection for SCH.

Fig 4 highlights the overall prevalence of STH by district. Prevalence varied significantly across districts, with seven districts reporting a prevalence of any STH ≥50%. The highest prevalence was observed in Rubavu (85.1%), followed by Rutsiro (71.0%) and Nyabihu (69.5%). The lowest prevalence of any STH was reported in Kicukiro (8.6%) and Gasabo (15.3%), both located in Kigali City. Detailed district-level prevalence and intensity data for each species are provided in S1 Table.

## Prevalence of any STH and *S. mansoni* by individual and household sociodemographic characteristics

Table 5 presents the prevalence of STH and *S. mansoni* (positive on either KK or POC-CCA) by individual and household sociodemographic characteristics. The prevalence of any STH varied by province, with higher rates in the West (58.5%)

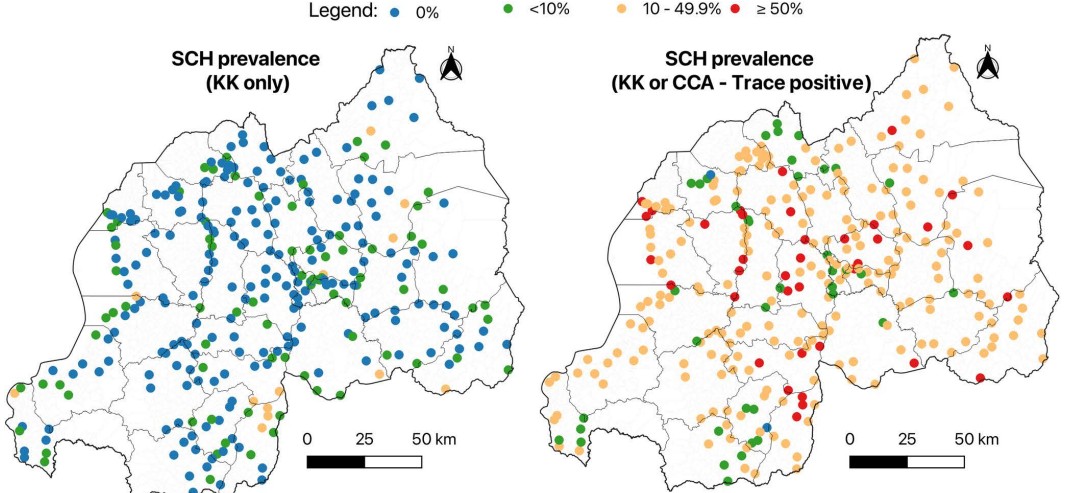

**Fig 3. Geographical distribution of S. *mansoni* prevalence in Rwanda (KK and POC-CCA, traces considered as positive)** The base layer consists of shapefiles provided by the National Institute of Statistics of Rwanda and publicly available through the Africa GeoPortal, powered by Esri: **https://rwanda.africageoportal.com/search?q=rwanda**. Notes: District-level mapping was not performed for S. mansoni as it would imply uniform prevalence and treatment eligibility across entire districts. Given the focal nature of transmission and the use of cells as the implementation unit in Rwanda, site-level data provide more actionable information for targeted intervention.

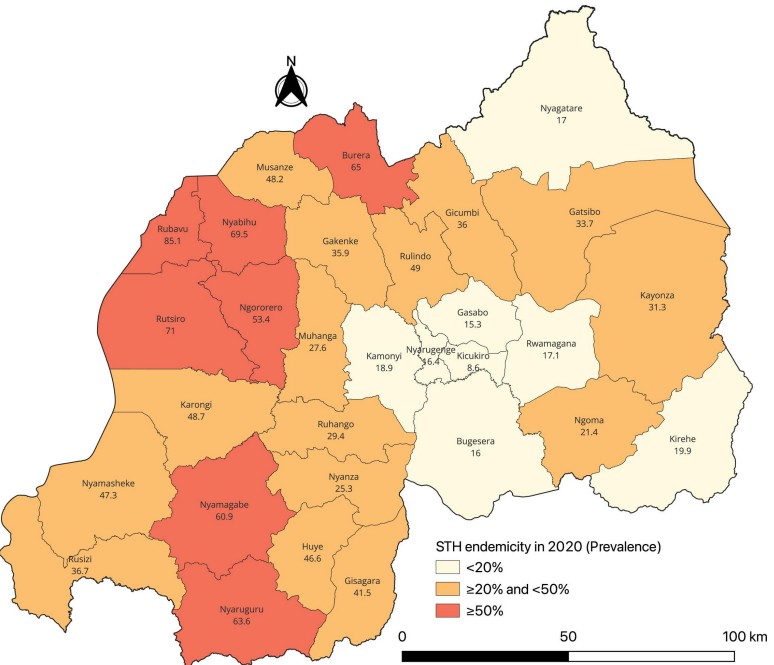

**Fig 4. Prevalence of any STH by district for all populations combined.** Numbers on the map represent district-level prevalence for any STH. The base layer consists of shapefiles provided by the National Institute of Statistics of Rwanda and publicly available through the Africa GeoPortal, powered by Esri: https://rwanda.africageoportal.com/search?q=rwanda.

and lower rates in Kigali City (13.5%) (p<0.001). Adults had a higher STH prevalence (46.1%) compared to pre-SAC at 30.2% and SAC at 38.8% (p<0.001). The prevalence was also higher among divorced, separated, or widowed individuals (51.8%, p<0.001). Individuals aged 10 years and above without formal education had the highest prevalence of STH (50.1%, p<0.001). Individuals engaged in agriculture/farming had higher rates of infection (48.4%), as well as those who reported being employed (formal/informal) (48.3%), compared to other occupational groups (p<0.001).

At the household level, STH prevalence was higher in homes lacking proper toilet facilities (61.9%) or with toilets less than 3 meters deep (62%) (p<0.001), as well as in households where soap and water were not consistently available (51.5%, p<0.001). Additionally, households relying on non-improved water sources or not treating water before drinking exhibited higher STH prevalence (50.6% and 51.5%, respectively, p<0.001).

For *S. mansoni*, in addition to higher prevalence among pre-SAC (35.4%) and lower among adults (20.9%) (p<0.001), differences were also seen by sex, with males showing higher prevalence (28.3%) compared to females (26.2%) (p=0.0017). Geographical factors played a role; prevalence was higher among those living near lakes (25.4%, p<0.001) and near rivers (18.6%, p=0.0315). Additionally, proximity to marshlands used for rice farming was associated with a higher prevalence (22.5%, p=0.0132).

Details on prevalence and intensity of Hookworm, *A. lumbricoides*, *T. trichiura*, and *S. mansoni* (KK, POC-CCA Trace Positive/Negative) by sociodemographic characteristics are presented in S2 Table. Univariate regression results, including odds ratios, 95% confidence intervals, and Chi-square test statistics, are presented in S3 Table.

### Change in prevalence of STH and *S. mansoni* from 2008 to 2020

Fig 5 shows changes in infection prevalence across surveys from 2008, 2014, and 2020. The prevalence of any STH declined significantly by 20.8% points from 2008 to 2014 (95% CI: -22.2, -19.4) and by 6.2 points from 2014 to 2020 (95% CI: -7.8, -4.6). *A. lumbricoides* prevalence fell by 1.6 points between 2008 and 2014 (95% CI: -3.0, -0.2) and by 6.5 percentage points from 2014 to 2020 (95% CI: -8.0, -5.0). Hookworm dropped sharply by 26.6 percentage points from 2008 to 2014 (95% CI: -27.7, -25.5) but slightly increased by 1.1 percentage points from 2014 to 2020, *T. trichiura* prevalence declined by 4 points from 2008 to 2014 (95% CI: -5.3, -2.7) and by 8 points from 2014 to 2020 (95% CI: -9.2, -6.8). *S. mansoni* prevalence using the CCA method (trace positive) dropped by 10.5 points from 2014 to 2020 (95% CI: -12.0, -9.0). Details of the estimates from previous surveys for each STH and SCH are available in the S4 Table.

Fig 6 shows the spatial distribution of STH endemicity in Rwanda in 2008, 2014, and 2020 among SAC. In 2008, 23 out of 30 districts (76.7%) had high STH prevalence (≥50%), with only 2 districts (6.7%) showing prevalence <20%. By 2014, high-prevalence districts decreased to 12 (40%). In 2020, high-prevalence districts further dropped to 10 (33.3%). Additionally, regarding the progress toward their elimination as the public health problem, fourteen of the 30 districts had an MHI prevalence of ≥2% for at least one STH species among SAC.

However, some districts experienced an increase in prevalence. Two districts (Kayonza and Ruhango), which had prevalence <20% in earlier years, shifted to the category of prevalence between ≥20% and <50% by 2020. Additionally, one district (Rulindo) showed an increase, shifting from prevalence between ≥20% and <50% to ≥50% in 2020 (Fig 6).

### Knowledge, attitudes, and practices about STH and SCH

Table 6 summarizes sources of information, knowledge, and attitudes about STH and SCH among SAC and adults. Knowledge was high, with 90.6% of SAC and 96.6% of adults aware of correct transmission modes, and over 90% understanding proper treatment. However, 25% believed prevention was impossible, and 12.7% found using untreated human waste as fertilizer acceptable. For STH, SAC mainly received information from schools (69.1%), while adults relied on health facilities (54.1%) and media (32.9%).

**Table 5. Prevalence of any STH and any *S. mansoni* (positive on either KK or POC-CCA) by individual and household sociodemographic characteristics.**

| Variables | Categories | | Any STH | | | S. *mansoni* (KK or POC-CCA - Trace positive) | | |
|---|---|---|---|---|---|---|---|---|
| | | | n (%) | Total (N) | P value | n (%) | Total (N) | P value |
| **Social demographics** | | | | | | | | |
| Province | East | | 908 (22.3) | 4068 | | 1227 (29.9) | 4109 | <0.001 |
| | Kigali City | | 223 (13.5) | 1649 | | 520 (29.6) | 1757 | |
| | North | | 1375 (46.9) | 2933 | | 748 (25.1) | 2976 | |
| | South | | 1826 (39.3) | 4645 | | 1195 (25.2) | 4733 | |
| | West | | 2380 (58.5) | 4065 | | 1129 (27.2) | 4144 | |
| Sex of the participant | Female | | 3655 (39.1) | 9352 | 0.2265 | 2505 (26.2) | 9553 | 0.0017 |
| | Male | | 3057 (38.2) | 8008 | | 2314 (28.3) | 8166 | |
| Age categories | Adults (16+) | | 2739 (46.1) | 5947 | <0.001 | 1260 (20.9) | 6018 | <0.001 |
| | pre-SAC (1 - 4) | | 1603 (30.2) | 5309 | | 1962 (35.4) | 5542 | |
| | SAC (5 - 15) | | 2370 (38.8) | 6104 | | 1597 (25.9) | 6159 | |
| Marital status | Single | | 1280 (41) | 3122 | <0.001 | 824 (26.2) | 3143 | <0.001 |
| | Married | | 2164 (45.2) | 4790 | | 977 (20.2) | 4847 | |
| | Divorced/Separated/Widowed | | 284 (51.8) | 548 | | 108 (19.4) | 557 | |
| Education level | No education | | 2300 (50.1) | 4589 | <0.001 | 1068 (23.1) | 4619 | 0.3151 |
| | Primary | | 1358 (38.4) | 3538 | | 767 (21.4) | 3585 | |
| | Vocational or Literacy training | | 5 (33.3) | 15 | | 3 (20) | 15 | |
| | Secondary and Higher | | 67 (20.8) | 322 | | 73 (22) | 332 | |
| Main occupation | Agriculture/Farmer | | 2398 (48.4) | 4959 | <0.001 | 1019 (20.4) | 4999 | <0.001 |
| | Employed (formal/informal) | | 188 (48.3) | 389 | | 86 (21.7) | 397 | |
| | Self Employed/Retired | | 30 (23.6) | 127 | | 38 (28.1) | 135 | |
| | Student | | 1044 (39.3) | 2659 | | 695 (26) | 2677 | |
| | Unemployed/Other | | 70 (21.1) | 331 | | 73 (21.2) | 344 | |
| *Household-level variables* | | | | | | | | |
| Has toilet facility | No | | 39 (61.9) | 63 | <0.001 | 15 (23.1) | 65 | 0.7708 |
| | Yes, < 3 meters deep | | 342 (62) | 552 | | 107 (19.3) | 554 | |
| | Yes, 3+ meters deep | | 1894 (44.7) | 4234 | | 846 (19.8) | 4281 | |
| Always have soap and water | No | | 1046 (51.5) | 2033 | <0.001 | 426 (20.9) | 2042 | 0.0921 |
| | Yes | | 1190 (43.2) | 2753 | | 527 (18.9) | 2793 | |
| Main water source | Improved source | | 781 (41.2) | 1897 | <0.001 | 384 (19.9) | 1929 | 0.8587 |
| | Non-improved source | | 1494 (50.6) | 2952 | | 584 (19.7) | 2971 | |
| Water treatment before drinking | Never | | 1129 (51.5) | 2192 | <0.001 | 426 (19.3) | 2208 | 0.6443 |
| | Sometimes | | 527 (47.7) | 1104 | | 229 (20.7) | 1108 | |
| | Yes, always | | 619 (39.9) | 1553 | | 313 (19.8) | 1584 | |
| | Water treatment method (among those who always treat water) | Boiling | 584 (40.4) | 1444 | 0.3334 | 296 (20.1) | 1471 | 0.6165 |
| | | Chemical | 9 (27.3) | 33 | | 5 (14.7) | 34 | |
| | | Filtering, | 21 (34.4) | 61 | | 9 (14.5) | 62 | |
| | | Others | 5 (33.3) | 15 | | 3 (17.6) | 17 | |
| Time to closest water source (round trip) | 30 - 60 min | | 448 (51.1) | 876 | 0.0663 | 162 (18.3) | 884 | 0.3679 |
| | More than 1 hour | | 310 (55.1) | 563 | | 112 (19.7) | 568 | |
| | On premises or <30 min | | 830 (49.4) | 1680 | | 349 (20.7) | 1689 | |

*(Continued)*

**Table 5.** (Continued)

| Variables | Categories | | Any STH | | | S. *mansoni* (KK or POC-CCA - Trace positive) | | |
|---|---|---|---|---|---|---|---|---|
| | | | n (%) | Total (N) | P value | n (%) | Total (N) | P value |
| Use human excreta as fertilizer and treat it | No | | 1516 (43.5) | 3486 | <0.001 | 662 (18.7) | 3533 | 0.0083 |
| | Yes, & treat the fertilizer used | | 597 (56.1) | 1064 | | 231 (21.7) | 1066 | |
| | Yes, does not treat fertilizer | | 123 (52.1) | 236 | | 60 (25.4) | 236 | |
| Nearest water bodies (N=4906) (Participants could have more than one type.) | Lake | No | 1709 (46.8) | 3652 | 0.7944 | 663 (17.9) | 3699 | <0.001 |
| | | Yes | 566 (47.3) | 1197 | | 305 (25.4) | 1201 | |
| | River | No | 922 (40.9) | 2253 | <0.001 | 482 (21.1) | 2286 | 0.0315 |
| | | Yes | 1353 (52.1) | 2596 | | 486 (18.6) | 2614 | |
| | Pond/Dam | No | 1729 (46.8) | 3697 | 0.7344 | 716 (19.2) | 3735 | 0.0719 |
| | | Yes | 546 (47.4) | 1152 | | 252 (21.6) | 1165 | |
| | Marshlands for rice farming | No | 1809 (47.8) | 3788 | 0.0294 | 728 (19) | 3832 | 0.0132 |
| | | Yes | 466 (43.9) | 1061 | | 240 (22.5) | 1068 | |
| | Marshlands for other | No | 765 (44.1) | 1733 | 0.0042 | 378 (21.6) | 1750 | 0.0173 |
| | | Yes | 1510 (48.5) | 3116 | | 590 (18.7) | 3150 | |
| | Other water bodies | No | 2181 (47.1) | 4631 | 0.2800 | 931 (19.9) | 4679 | 0.2869 |
| | | Yes | 94 (43.1) | 218 | | 37 (16.7) | 221 | |
| Walk time to closest water bodies | 0-20 min | | 1335 (47) | 2841 | 0.2444 | 562 (19.6) | 2868 | 0.9454 |
| | 21-40 min | | 656 (48.1) | 1364 | | 277 (20) | 1385 | |
| | 41+ min | | 284 (44.1) | 644 | | 129 (19.9) | 647 | |

The knowledge about SCH transmission was lower, with 52.6% of SAC and 50.1% of adults aware of the correct transmission modes. SAC primarily learned from schools (36.8%) and media (30.3%), while adults accessed information from media (45.7%) and health facilities (38.4%). Still, there was strong support for regular screening (93.2%) and medication (93%), though 36.7% viewed SCH as a mild disease.

## Risk factors for any STH and SCH: results from Mixed-Effect Logistic Regression Models

Table 7 shows the results from mixed effect logistic regression models identifying factors associated with STH and *S. mansoni* (SCH) infections, adjusted for other variables and accounting for variations among districts.

Several sociodemographic, environmental, and behavioral factors were associated with STH infection. Single individuals had significantly higher odds of STH infection compared to those who were married (AOR: 1.74, 95% CI: 1.32–2.28, p < 0.001). Those with no formal education were more likely to be infected compared to individuals with primary education (AOR: 1.56, 95% CI: 1.37–1.78, p < 0.001). Use of treated human excreta as fertilizer was associated with increased odds of STH infection (AOR: 1.43, 95% CI: 1.22–1.66, p < 0.001), as was the use of non-improved water sources (AOR: 1.17, 95% CI: 1.02–1.34, p = 0.024). Additionally, residing near marshlands used for other activities was linked to higher odds of infection (AOR: 1.17, 95% CI: 1.02–1.35, p = 0.025).

In contrast, several protective factors were identified. Individuals with secondary or higher education had significantly lower odds of infection compared to those with primary education (AOR: 0.55, 95% CI: 0.37–0.81, p = 0.003). Unemployed individuals (AOR: 0.34, 95% CI: 0.23–0.51, p < 0.001) and those who were self-employed or retired (AOR: 0.53, 95% CI: 0.31–0.91, p = 0.020) had lower odds of infection compared to those working in agriculture or farming. Households with toilet facilities deeper than three meters had reduced odds of STH infection (AOR: 0.78, 95% CI: 0.64–0.96, p = 0.018), and consistent treatment of drinking water was also associated with a protective effect (AOR: 0.79, 95% CI: 0.68–0.92, p = 0.003).

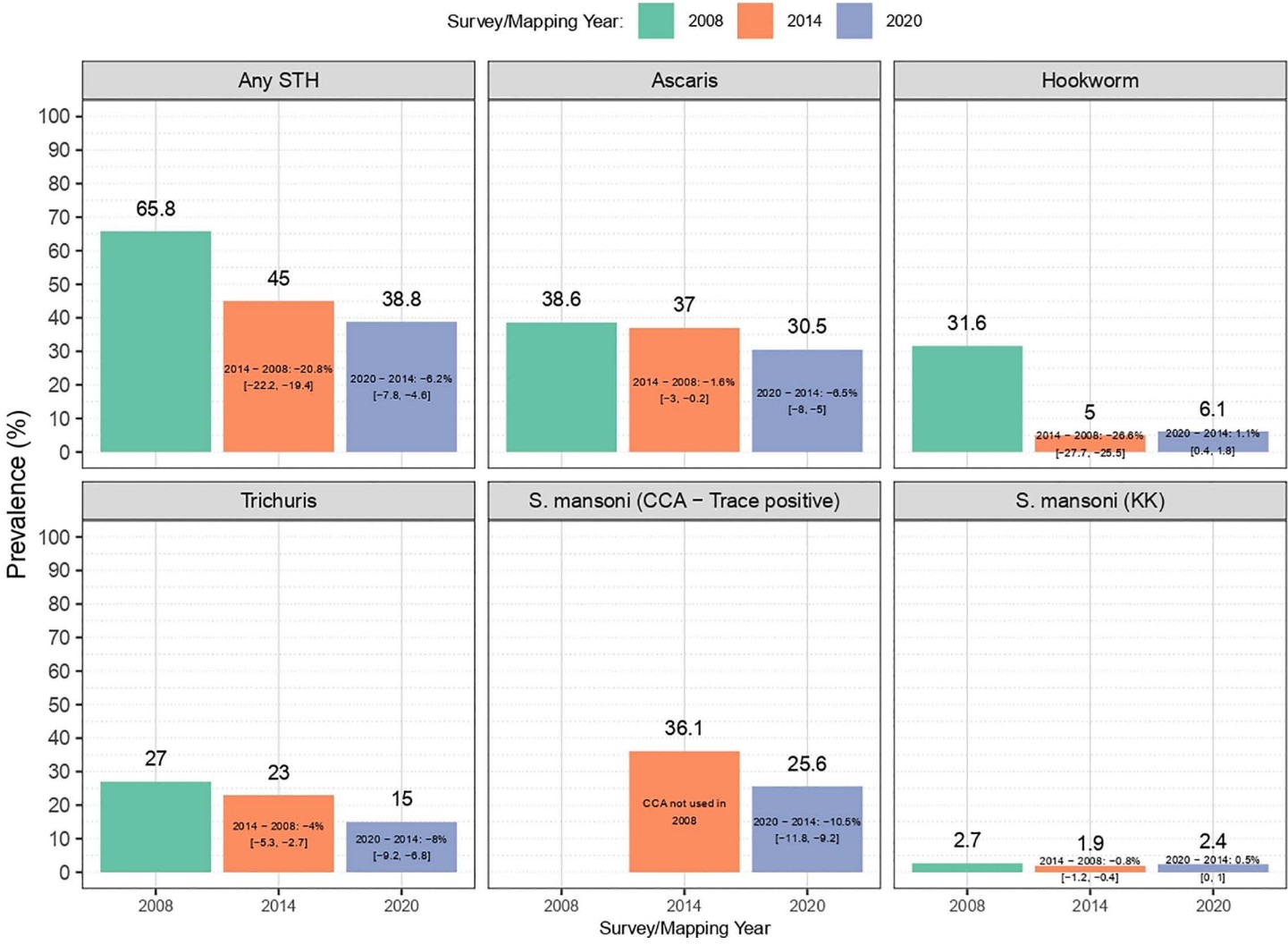

**Fig 5. Comparison of SCH and STH survey prevalence among SAC across mapping years.** Estimates represent absolute differences in preva-lence between surveys with corresponding 95% confidence intervals. Data include SAC aged 10–16 years in 2008, 9–18 years in 2014, and 5–15 years in 2020.

For SCH, single individuals were more likely to be infected compared to those who were married (AOR: 1.61, 95% CI: 1.11–2.33, p = 0.011). Participants with no formal education had higher odds of infection than those with primary education (AOR: 1.41, 95% CI: 1.16–1.72, p = 0.001). Living more than one hour from a water source was associated with increased odds of SCH infection compared to living 30–60 minutes away (AOR: 1.42, 95% CI: 1.06–1.91, p = 0.019). Proximity to lakes was also a significant risk factor (AOR: 1.76, 95% CI: 1.37–2.27, p < 0.001).

Use of human excreta as manure—whether untreated (AOR: 1.60, 95% CI: 1.06–2.41, p = 0.027) or treated before use (AOR: 1.32, 95% CI: 1.06–1.65, p = 0.014)—was linked to higher odds of SCH infection. Similarly, residing near marshlands used for rice farming was associated with increased infection risk (AOR: 1.31, 95% CI: 1.00–1.72, p = 0.047).

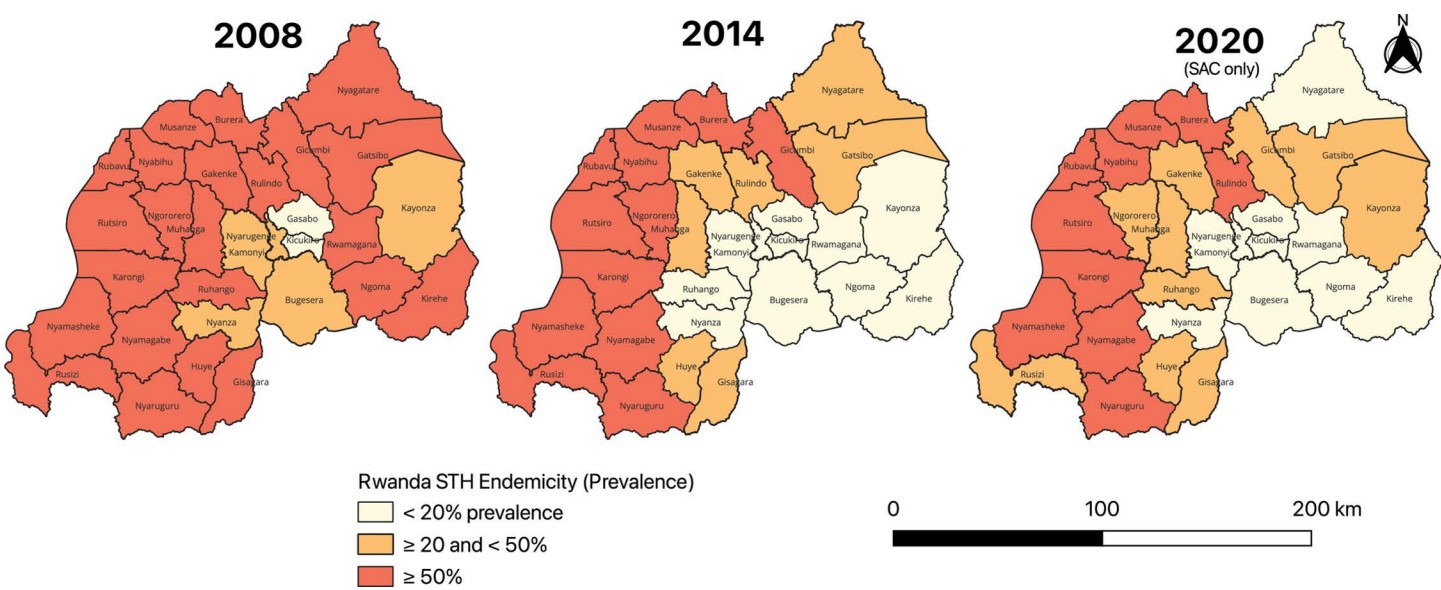

**Fig 6. Trends in STH endemicity among SAC across Rwandan districts (2008–2020).** Data include SAC aged 10–16 years in 2008, 9–18 years in 2014, and 5–15 years in 2020. The base layer consists of shapefiles provided by the National Institute of Statistics of Rwanda and publicly available through the Africa GeoPortal, powered by Esri: https://rwanda.africageoportal.com/search?q=rwanda.

## Discussion

### Key results and interpretation

This nationwide survey provides an updated assessment of the prevalence and risk factors for STH and SCH across all age groups in Rwanda. STH prevalence remains high, with adults showing the highest rates and pre-SAC the lowest. Fourteen districts reported an MHI prevalence of ≥2% for at least one STH species among school-aged children, highlighting persistent challenges in achieving the threshold for elimination as a public health problem. Similarly, SCH prevalence remains significant, particularly when assessed using POC-CCA diagnostics, which reveal prevalence exceeding 10% at several sites, necessitating preventive chemotherapy. Despite this, heavy intensity SCH infections remain below 1% across all endemic units, suggesting progress toward elimination targets.

A higher likelihood of STH infection was associated with being single, having no education, using treated human excreta as manure, relying on unimproved water sources, and living near marshlands. Lower likelihoods were observed among those with higher education, unemployed or self-employed/retired individuals, households with deep toilet facilities, and those practicing consistent water treatment. Similarly, SCH infection risk was higher among single individuals, lacked education, lived near lakes or rice-farming marshlands, used untreated or treated human excreta as fertilizer, or resided over an hour from a water source. These results underscore persistent vulnerabilities and highlight areas for targeted interventions to reduce the burden of STH and SCH in Rwanda.

***Persistence of high prevalence despite consistent MDA.*** Despite multiple rounds of high-coverage MDA, STH prevalence remains substantial in Rwanda. The study took place after 12 years of MDs implementation with 24 MDA rounds (two rounds per year) for STH and 12 rounds for SCH (one round annually) and maintained recommended effective MDA treatment coverage (>75%). This persistence can be attributed to several key factors, including limited access to clean water and sanitation, the exclusion of adults from MDA programs, and the limited efficacy of MDA drugs against certain helminths, particularly *T. trichiura*.

**Table 6. Source of information, knowledge, and attitudes about SCH and STH.**

| Variables | Categories | About SCH | | | About STH | | |
|---|---|---|---|---|---|---|---|
| | | SAC** (N = 152) | Adults (N = 1382) | Total (N = 1534) | SAC** (N = 1471) | Adults (N = 4791) | Total (N = 6262) |
| | | n (%) | | | n (%) | | |
| Source of information | Churches | 1 (0.7) | 6 (0.4) | 7 (0.5) | 9 (0.6) | 41 (0.9) | 50 (0.8) |
| | Community Health Workers | 36 (23.7) | 471 (34.1) | 507 (33.1) | | | |
| | Health facility | 44 (28.9) | 531 (38.4) | 575 (37.5) | 369 (25.1) | 2594 (54.1) | 2963 (47.3) |
| | Media | 46 (30.3) | 631 (45.7) | 677 (44.1) | 260 (17.7) | 1575 (32.9) | 1835 (29.3) |
| | School | 56 (36.8) | 88 (6.4) | 144 (9.4) | 1017 (69.1) | 364 (7.6) | 1381 (22.1) |
| | Old parents | 20 (13.2) | 67 (4.8) | 87 (5.7) | 319 (21.7) | 324 (6.8) | 643 (10.3) |
| | Other | 4 (2.6) | 29 (2.1) | 33 (2.2) | 12 (0.8) | 93 (1.9) | 105 (1.7) |
| When last received STH information | 1 year or less | 61 (40.1) | 439 (31.8) | 500 (32.6) | | | |
| | > 1 year | 91 (59.9) | 943 (68.2) | 1034 (67.4) | | | |
| Knows correct transmission mode | | 80 (52.6) | 692 (50.1) | 772 (50.3) | 1333 (90.6) | 4630 (96.6) | 5963 (95.2) |
| Know correct treatment | | * | | | 1344 (91.4) | 4549 (94.9) | 5893 (94.1) |
| Participant agreed that: | SCH is a mild disease. | 70 (46.1) | 493 (35.7) | 563 (36.7) | * | | |
| | Regular screening for SCH is important. | 134 (88.2) | 1296 (93.8) | 1430 (93.2) | * | | |
| | Taking SCH medication is important. | 136 (89.5) | 1290 (93.3) | 1426 (93) | | | |
| | Seeking care for symptoms like blood in stool or abdominal pain is necessary. | 143 (94.1) | 1358 (98.3) | 1501 (97.8) | * | | |
| | Intestinal worms cannot be prevented. | * | | | 363 (24.7) | 1205 (25.2) | 1568 (25) |
| | Using untreated human waste as fertilizer is acceptable. | * | | | 181 (12.3) | 614 (12.8) | 795 (12.7) |
| | Herbs are more effective than modern medicine for treating intestinal worms. | * | | | 213 (14.5) | 486 (10.1) | 699 (11.2) |

*Not assessed for this condition

**SAC were children aged 10–15 years, the age group able to answer knowledge questions.

A major barrier to effective control of STH and SCH is the limited implementation of comprehensive Water, Sanitation, and Hygiene (WASH) measures to support MDA. Despite Rwanda's investments in hygiene and sanitation infrastructure, progress has been slow. From 2014 to 2020, the population using improved water sources increased only modestly, from 72.4% to 79.6%, while households with sanitation facilities declined slightly, from 71.2% to 70.2% [13,14]. Additionally, data on the quality, maintenance, and consistent use of WASH facilities remain limited. This stagnation suggests that high MDA coverage in SAC alone may be insufficient to interrupt transmission, especially in communities with inconsistent access to clean water and sanitation [42]. Findings from this study underscore the need for WASH interventions that extend beyond MDA, emphasizing not only the availability of facilities but also their quality, regular maintenance, and reliable usage. Prioritizing these factors is essential for sustained control of STH and SCH transmission. Though requiring substantial investment and stakeholder collaboration, the long-term benefits are likely sustainable and impactful for infection control [43,44].

In addition to WASH and infrastructure gaps, the inclusion of adults in our survey revealed that adults have a higher STH prevalence (46.1%) compared to pre-SAC (30.2%). This finding aligns with studies from Benin, Malawi, Kenya, and India, highlighting those untreated adults may act as reservoirs for reinfection, undermining MDA efforts focused on children [15,16,45,46]. The higher prevalence in adults appears to be driven by elevated hookworm infections in this group and a notable prevalence of *A. lumbricoides* across all age groups. Historically, MDA in Rwanda has targeted pre-SAC and SAC as they are the most accessible and vulnerable populations. However, excluding adults leaves a substantial portion of the population untreated, enabling infections to persist and reinfect younger populations [47–49]. Expanding MDA

**Table 7. Results from the mixed effect logistic regression for the factors associated with having any STH, and the factor associated with _S. mansoni_.**

| Variables | Categories | | Any STH | | | _S. mansoni_ (KK or POC-CCA - Trace positive) | | |
|---|---|---|---|---|---|---|---|---|
| | | | AOR | 95% CI | P-value | AOR | 95% CI | P-value |
| Age categories | Adults (16+) | | Ref | | | Ref | | |
| | SAC (5–15) | | 1.3 | (0.53 - 3.16) | 0.565 | 1.79 | (0.61 - 5.24) | 0.285 |
| Marital status | Married | | Ref | | | Ref | | |
| | Single | | 1.74 | (1.32 - 2.28) | <0.001* | 1.61 | (1.11 - 2.33) | 0.011* |
| | Divorced/Separated/Widowed | | 1.42 | (1.16 - 1.74) | 0.001* | 0.83 | (0.6 - 1.14) | 0.251 |
| Education level | Primary | | Ref | | | Ref | | |
| | No education | | 1.56 | (1.37 - 1.78) | <0.001* | 1.41 | (1.16 - 1.72) | 0.001* |
| | Vocational or literacy training | | 0.68 | (0.19 - 2.42) | 0.554 | 1.27 | (0.14 - 11.34) | 0.830 |
| | Secondary and higher | | 0.55 | (0.37 - 0.81) | 0.003* | 0.8 | (0.41 - 1.58) | 0.526 |
| Main occupation | Agriculture/Farmer | | Ref | | | Ref | | |
| | Employed (formal/informal) | | 1.03 | (0.78 - 1.37) | 0.829 | 1.1 | (0.73 - 1.65) | 0.663 |
| | Self Employed/Retired | | 0.53 | (0.31 - 0.91) | 0.020* | 1.06 | (0.48 - 2.37) | 0.880 |
| | Student | | 0.54 | (0.34 - 0.85) | 0.008* | 1.26 | (0.68 - 2.35) | 0.463 |
| | Unemployed/Other | | 0.34 | (0.23 - 0.51) | <0.001* | 0.82 | (0.44 - 1.54) | 0.540 |
| Has toilet facility | Yes, <3 meters deep | | Ref | | | | | |
| | Yes, 3+ meters deep | | 0.78 | (0.64 - 0.96) | 0.018* | | | |
| Always have soap and water | No | | Ref | | | | | |
| | Yes | | 0.95 | (0.84 - 1.08) | 0.460 | | | |
| Main water source | Improved source | | Ref | | | Ref | | |
| | Non-improved source | | 1.17 | (1.02 - 1.34) | 0.024* | 0.71 | (0.47 - 1.07) | 0.099 |
| Treat water before drinking | Never | | Ref | | | | | |
| | Sometimes | | 0.84 | (0.71 - 0.98) | 0.027* | | | |
| | Yes, always | | 0.79 | (0.68 - 0.92) | 0.003* | | | |
| Time to closest water source (round trip) | 30 - 60 min | | | | | Ref | | |
| | More than 1 hour | | | | | 1.42 | (1.06 - 1.91) | 0.019* |
| | On premises or <30 min | | | | | 1.17 | (0.91 - 1.49) | 0.215 |
| Use human excreta as fertilizer and treat it | No | | Ref | | | Ref | | |
| | Yes, and treat the fertilizer used | | 1.43 | (1.22 - 1.66) | <0.001* | 1.32 | (1.06 - 1.65) | 0.014* |
| | Yes, does not treat fertilizer used | | 1.19 | (0.9 - 1.59) | 0.228 | 1.6 | (1.06 - 2.41) | 0.027* |
| Nearest water bodies | Lake | No | | | | Ref | | |
| | | Yes | | | | 1.76 | (1.37 - 2.27) | <0.001* |
| | River | No | | | | Ref | | |
| | | Yes | | | | 0.89 | (0.7 - 1.12) | 0.305 |
| | Pond/Dam | No | | | | Ref | | |
| | | Yes | | | | 0.96 | (0.76 - 1.22) | 0.748 |
| | Marshlands for rice farming | No | Ref | | | Ref | | |
| | | Yes | 1.02 | (0.87 - 1.21) | 0.777 | 1.31 | (1 - 1.72) | 0.047* |
| | Marshlands for other | No | Ref | | | Ref | | |
| | | Yes | 1.17 | (1.02 - 1.35) | 0.025* | 0.85 | (0.68 - 1.07) | 0.168 |
| | Other water bodies | No | | | | Ref | | |
| | | Yes | | | | 0.98 | (0.56 - 1.73) | 0.948 |

_(Continued)_

**Table 7.** (Continued)

| Variables | Categories | Any STH | | | S. mansoni (KK or POC-CCA - Trace positive) | | |
|---|---|---|---|---|---|---|---|
| | | AOR | 95% CI | P-value | AOR | 95% CI | P-value |
| Walk time to closest water bodies | 0-20 min | | | | Ref | | |
| | 21-40 min | | | | 1.04 | (0.83 - 1.31) | 0.728 |
| | 41 + min | | | | 0.9 | (0.67 - 1.23) | 0.518 |
| Standard deviation for the random intercept (Districts) | | 0.6 | | | 0.73 | | |

*Model limited to age >=10 years. Empty categories were not included in the model for the specific condition.*

*\* Significant variables (p value <0.05)*

to include adults could more effectively disrupt transmission and accelerate progress toward eliminating STH and SCH as public health problems [50].

However, including adults in MDA efforts presents logistical and financial challenges. The health benefits of switching to community-wide treatment depend on the STH species and baseline endemicity [51,52]. More importantly, WHO does not currently support drug donations for adult STH treatment in MDA programs, so expansion would require additional funding—a significant cost for the government, given the scale needed [51,53,54]. While advocating for donor support to include adults, a phased approach may be effective, focusing on high-prevalence districts (e.g., ≥ 50% prevalence) where adult treatment could have the greatest impact. Targeting these areas first would help the government optimize resources while addressing the most urgent needs.

In contrast, alongside gaps in MDA coverage, the limited effectiveness of albendazole against *T. trichiura* remains a major challenge for STH control. Albendazole has demonstrated low cure rates for *T. trichiura*, whereas combination therapy with ivermectin has shown higher egg reduction and cure rates [55]. Findings from this study, supported by results in S1 and S2 Tables, indicate persistently high STH prevalence in areas with substantial *T. trichiura* burden, likely due to the limited efficacy of albendazole. These results suggest that implementing combination therapies, such as albendazole with ivermectin, in *T. trichiura*-endemic areas could enhance MDA effectiveness and accelerate progress toward STH control and elimination targets.

Our study observed a more pronounced decline in hookworm prevalence compared to Ascaris lumbricoides and *T. trichiura*, likely attributable to albendazole's higher efficacy against hookworm and concurrent behavioral and environmental improvements. National hygiene initiatives such as mandatory shoe-wearing policies, particularly among school-aged children and reductions in open defecation may have disproportionately disrupted hookworm transmission. Nevertheless, hookworm infection remains more prevalent among adults, likely due to continued exposure through bare-foot agricultural activities. These findings underscore the need to expand behavior change interventions beyond school settings to effectively reach adult populations, with targeted messaging promoting consistent footwear use and other protective practices.

[55] We acknowledge that our purposive, risk-based sampling strategy designed to identify high-transmission areas near water bodies may have resulted in higher prevalence estimates than would be obtained through representative sampling. While this targeted approach enhances programmatic relevance by focusing on areas most likely to require intervention, it may also overestimate the true burden of disease at broader geographic levels and thereby underestimate the impact of previous treatment efforts. For example, the SCH prevalence based on KK in this survey was 2.4%, compared to 1.9% reported in the 2014 national mapping. However, this comparison should be interpreted with caution, as the 2014 survey employed school-based sampling with wider geographic coverage and did not specifically target high-risk areas, limiting direct comparability. Importantly, as Rwanda moves toward interruption of transmission, focusing on the

highest-risk areas ensures that control efforts are directed where they are most needed, and provides a conservative upper bound for prevalence, offering assurance that transmission levels in lower-risk areas are likely to be even lower.

The POC-CCA test provided a more sensitive measure of *S. mansoni* prevalence in Rwanda than the KK method, particularly for detecting light and asymptomatic infections in a context of declining transmission due to ongoing mass drug administration. Including trace results raised prevalence to 27.2%, compared to 1.7% with KK, offering a more comprehensive picture of ongoing transmission risks. Although trace results remain debated [10,11], their inclusion aligns with Rwanda's elimination goals, especially given the availability of praziquantel to support expanded treatment. These findings underscore the value of POC-CCA in low-endemicity settings while highlighting the importance of using both diagnostics during the transition toward elimination, given that current WHO guidelines still rely on KK for assessing elimination as a public health problem.

**Factors associated with STH and SCH infection.** Our findings highlight how socioeconomic, occupational, and environmental factors significantly influence the likelihood of STH and SCH infections, underscoring the need for targeted interventions to reduce transmission.

Socioeconomic factors such as being single, education, water source quality, sanitation practices, and employment status emerged as key determinants of infection likelihood, similar to findings in Benin, India, Ghana, and Ethiopia, where lower-income groups face higher infection likelihood [45,46,56–58]. The association between being single and increased odds of STH infection may reflect socio-behavioral and occupational patterns, as single individuals, particularly adult males, may be more engaged in farming or manual labor, which increases exposure to contaminated soil. They may also be less likely to participate in household-based MDA or seek preventive care. Similarly, education is often correlated with economic status and better access to sanitation, reducing contact with contaminated materials. Individuals without formal education are more susceptible to infection, likely due to limited awareness of preventive practices like hand-washing and safe food handling, along with restricted access to clean water and sanitation facilities. In contrast, those with higher education are more likely to adopt preventive behaviors, such as regular hand-washing and water treatment, which reduce infection risk [59].

Furthermore, reliance on unimproved water sources and the use of human excreta as manure were associated with a higher likelihood of infection. In lower-income communities, these practices often result from a lack of affordable, safer alternatives, increasing exposure to contaminants [56,60,61]. Households using treated human waste as manure may mistakenly assume it is safe, often relying on limited or inadequate protective measures and equipment. In some cases, they may adopt certain protective practices but wrongly believe these are sufficient, even when they are not effective. Expanding access to improved sanitation facilities, clean water sources, and modern fertilizers, along with raising awareness about the health risks associated with using human excreta, can promote safer waste disposal and hygiene practices, reducing contact with infectious agents and lowering infection rates.

Beyond socioeconomic factors, occupational and environmental exposures significantly influenced infection risk. Specifically, residing near marshlands, lakes, or rice-farming areas was strongly associated with higher infection rates, as these environments facilitate SCH transmission through snail vectors and support the survival of STH eggs and larvae [43]. Proximity to such water bodies increases exposure to contaminated water and soil, heightening the risk of infection. Evidence shows that adequate sanitation is associated with a lower likelihood of *S. mansoni* infection [62]. In this study, participants living farther from clean water sources had higher SCH prevalence, likely relying on untreated local water bodies for daily needs, which prolongs exposure to contamination [42]. These findings underscore the importance of context-specific interventions, including improving access to clean water and sanitation in high-risk areas.

In addition to environmental risks, occupational exposures play a critical role, particularly for individuals in jobs involving frequent contact with soil or water, such as farming and manual labor. These occupations lead to ongoing exposure

to environments where STH eggs and SCH larvae thrive, making infection difficult to avoid without protective equipment or adequate sanitation facilities [43]. For example, rice farming, which requires prolonged water exposure, substantially increases infection risk [63–65]. Some studies have also demonstrated a correlation between helminth egg counts in wastewater and soil samples and infections in farmers [66]. Our study suggests that the persistence of hookworm among adults (21% prevalence compared to 4.3% and 6.1% among pre-SAC and SAC) may be linked to occupational and rural lifestyle factors, compounded by the exclusion of adults from MDA programs. These findings support the need for control strategies that extend beyond MDA, including promoting consistent shoe-wearing, improving WASH infrastructure, and providing targeted hygiene education to reduce infection risks and support efforts to lower morbidity in high-risk communities [1,2,62].

### Strengths and limitations

This study is the first national, community-based survey of STH and SCH in Rwanda, providing a broader perspective compared to traditional school-based assessments. By including adults alongside the traditionally assessed pre-SAC and SAC participants, it offers a comprehensive view of infection prevalence across all age groups. A key strength of the study was the use of focused mapping, which, while not fully aligned with the most recent WHO definition of precision mapping [24], emphasized targeting high-risk areas informed by ecological and local programmatic insights. This approach improved the accuracy of district-level estimates and ensured relevance for program planning by concentrating data collection where transmission was most likely. Furthermore, the combined use of KK and POC-CCA methods enhanced SCH detection, identifying cases that might have been missed by KK stool testing alone. Rigorous quality assurance measures, including training, pre-testing, and daily checks, ensured the collection of reliable and high-quality data.

Despite its strengths, the study has some limitations. The reliance on self-reported data for some variables, such as sanitation practices, could introduce recall bias. The use of a sampling strategy originally designed for SCH may have influenced STH estimates, as some low-risk villages were not included in the sampling frame; however, this limitation was likely mitigated by the large sample size and the broad geographic distribution of selected villages. Additionally, while the POC-CCA test offers greater sensitivity, particularly in low-endemic settings, it has batch-to-batch variations and may overestimate SCH prevalence due to false positives from trace readings, warranting further research to improve the accuracy of this diagnostic [10,11]. Future research should consider the WHO Target Product Profiles for schistosomiasis point-of-care products to address these challenges, focusing on developing reliable tools for low-prevalence areas. These TPPs aim to guide the creation of diagnostics that can accurately measure whether prevalence is above or below the 10% threshold, which is critical for determining MDA frequency [67]. Finally, another key limitation is that the study was powered to provide estimates only at the district level, not at the sub-district (e.g., cell or sector) level, which limits its direct utility for micro-targeting of MDA interventions. Future research should adopt precision assessment approaches and employ sampling designs that generate sub-district-level estimates to better inform localized treatment decisions.

### Conclusion

This nationwide survey highlights that, despite ongoing MDA programs, STH and SCH remain significant public health challenges in Rwanda, particularly among adults and in high-risk regions. Structural and behavioral factors, such as limited education, the use of untreated human waste as manure, and proximity to lakes and marshlands, sustain transmission and reveal the limitations of traditional, school-based approaches. To effectively address these infections, MDA programs must be extended to include adults and integrated with efforts to improve sanitation, access to clean water, and community engagement in safe hygiene practices. Tackling these drivers and social determinants will enable more equitable and sustainable control, advancing toward the elimination of these infections.

## Supporting information

**S1 Text.  Age-Adjusted Prevalence of Soil-Transmitted Helminths and Schistosoma mansoni.**
(DOCX)

**S1 Table.  District level Prevalence and intensity by species.**
(XLSX)

**S2 Table.  Prevalence of Hookworm, *A. lumbricoides*, *T. trichiura*, and *S. mansoni* (KK, CCA Trace Positive/Negative) by sociodemographic characteristics.**
(XLSX)

**S3 Table.  Univariate regression results for any STH and *Schistosoma mansoni* (ORs, 95% CIs, and Chi-Square Statistics).**
(XLSX)

**S4 Table.  Summary results of 2008, 2014, and 2020 surveys used for comparison.**
(DOCX)

**S1 Data.   Collection tools.**
(DOC)

**S1 File.  STROBE Checklist [68].**
(DOCX)

## Acknowledgments

We would like to express our sincere gratitude to all the participants who took part in this study, as well as the dedicated data collectors and field teams whose hard work made this research possible. We also appreciate the support of community health workers, local leaders, and health center staff who facilitated the smooth coordination of data collection across the districts.

## Author contributions

**Conceptualization:** Ladislas Nshimiyimana, Jean Bosco Mbonigaba, Aimable Mbituyumuremyi, Leonard Uwayezu, Joseph Kagabo, Elias Niyituma, Nadine Rujeni, Eugene Ruberanziza.

**Data curation:** Ladislas Nshimiyimana, Jean Bosco Mbonigaba, Dieudonne Hakizimana.

**Formal analysis:** Ladislas Nshimiyimana, Jean Bosco Mbonigaba, Dieudonne Hakizimana.

**Funding acquisition:** Jean Bosco Mbonigaba, Aimable Mbituyumuremyi.

**Investigation:** Ladislas Nshimiyimana, Jean Bosco Mbonigaba, Aimable Mbituyumuremyi, Leonard Uwayezu, Michee Kabera, Emmanuel Hakizimana, Phocas Mazimpaka, Emmanuel Ruzindana, Eliah Shema, Tharcisse Munyaneza, Jean Bosco Mucaca, Maurice Twahirwa, Esperance Umumararungu, Joseph Kagabo, Richard Habimana, Elias Niyituma, Eugene Ruberanziza.

**Methodology:** Ladislas Nshimiyimana, Jean Bosco Mbonigaba, Aimable Mbituyumuremyi, Elias Nyandwi, Leonard Uwayezu, Elias Niyituma, Nadine Rujeni, Eugene Ruberanziza.

**Project administration:** Jean Bosco Mbonigaba, Aimable Mbituyumuremyi, Karen Palacio.

**Resources:** Jean Bosco Mbonigaba, Aimable Mbituyumuremyi, Karen Palacio.

**Supervision:** Aimable Mbituyumuremyi, Alison Ower, Karen Palacio, Emmanuel Hakizimana, Eugene Ruberanziza.

**Visualization:** Ladislas Nshimiyimana, Jean Bosco Mbonigaba, Alison Ower, Dieudonne Hakizimana, Elias Nyandwi, Alphonse Mutabazi, Jeanne Uwizeyimana, Eugene Ruberanziza.

**Writing – original draft:** Ladislas Nshimiyimana, Alison Ower, Dieudonne Hakizimana, Eugene Ruberanziza.

**Writing – review & editing:** Ladislas Nshimiyimana, Jean Bosco Mbonigaba, Aimable Mbituyumuremyi, Alison Ower, Dieudonne Hakizimana, Elias Nyandwi, Karen Palacio, Alphonse Mutabazi, Jeanne Uwizeyimana, Leonard Uwayezu, Michee Kabera, Emmanuel Hakizimana, Phocas Mazimpaka, Emmanuel Ruzindana, Eliah Shema, Tharcisse Munyaneza, Jean Bosco Mucaca, Maurice Twahirwa, Esperance Umumararungu, Joseph Kagabo, Richard Habimana, Elias Niyituma, Tonya Huston, Jamie Tallant, Warren Lancaster, Nadine Rujeni, Eugene Ruberanziza.

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
