## [Decision Letter · Decision Letter 0]

10 Mar 2025

Remapping parasite landscapes: a nationwide precision mapping of Schistosomiasis and Soil-Transmitted Helminthiasis and associated risk factors in Rwanda

Dear Dr. NSHIMIYIMANA,

Thank you for submitting your manuscript to PLOS Neglected Tropical Diseases. After careful consideration, we feel that it has merit but does not fully meet PLOS Neglected Tropical Diseases's publication criteria as it currently stands. Therefore, we invite you to submit a revised version of the manuscript that addresses the points raised during the review process.

Please submit your revised manuscript within 60 days, byMay 7th. If you will need more time than this to complete your revisions, please reply to this message or contact the journal office at plosntds@plos.org. Please include the following items when submitting your revised manuscript:

We look forward to receiving your revised manuscript.

Kind regards,

Angela Monica Ionica, Ph.D.

Academic Editor

Jong-Yil Chai

Section Editor

Shaden Kamhawi

co-Editor-in-Chief

Paul Brindley

co-Editor-in-Chief

**Journal Requirements:**

At this stage, the following Authors/Authors require contributions: Ladislas Nshimiyimana, Aimable Mbituyumuremyi, Alison Ower, Dieudonne Hakizimana, Jean Bosco Mbonigaba, Elias Nyandwi, Karen Palacio, Alphonse Mutabazi, Jeanne Uwizeyimana, Leonard Uwayezu, Michee Kabera, Emmanuel Hakizimana, Phocas Mazimpaka, Emmanuel Ruzindana, Eliah Shema, Tharcisse Munyaneza, Jean Bosco Mucaca, Maurice Twahirwa, Esperance Umumararungu, Joseph Kagabo, Richard Habimana, Elias Niyituma, Tonya Huston, Jamie Tallant, Warren Lancaster, Nadine Rujeni, and Eugene Ruberanziza. Please ensure that the full contributions of each author are acknowledged in the "Add/Edit/Remove Authors" section of our submission form.

2) Please ensure that the funders and grant numbers match between the Financial Disclosure field and the Funding Information tab in your submission form. Note that the funders must be provided in the same order in both places as well.

**Reviewers' Comments:**

Reviewer's Responses to Questions

**Key Review Criteria Required for Acceptance?**

**Methods:**

-Are the objectives of the study clearly articulated with a clear testable hypothesis stated?

-Is the study design appropriate to address the stated objectives?

-Is the population clearly described and appropriate for the hypothesis being tested?

-Is the sample size sufficient to ensure adequate power to address the hypothesis being tested?

-Were correct statistical analysis used to support conclusions?

-Are there concerns about ethical or regulatory requirements being met?

Reviewer #1: (No Response)

Reviewer #2: Line 96 (Introduction): No description of the disease aside from the various worms causing the infection. Suggest providing at least a brief description of the mode of transmission and public health burden of SCH and STH, including e.g. the influence of sanitation, proximity to water bodies, any chronic conditions, risk of anaemia.

Line 191-193: What do trace results mean in relation to POC-CCA results? Perhaps a brief description of the test would aid readers in understanding how the test is run and the utility (sensitivity, specificity etc.) of POC-CCA for urine testing, especially in the field, instead of standard diagnosis by microscopic examination/urine filtration (noted in lines 255-256 that additional urine tests were only done when haematuria was detected).

Also, how would the POC-CCA results be compared to the previous 2 rounds of mapping? How were the urine samples tested previously?

Based these 3 lines, the authors presented SCH status when either stool or urine tested positive i.e.

1. POC-CCA trace results were negative

2. KK or POC-CCA tests were positive but trace results were negative

Reviewer #3: (No Response)

**Results:**

-Does the analysis presented match the analysis plan?

-Are the results clearly and completely presented?

-Are the figures (Tables, Images) of sufficient quality for clarity?

Reviewer #1: (No Response)

Reviewer #2: Line 506: S. mansoni prevalence – is there a prevalence map by district available to illustrate this?

Line 547: The AORs were adjusted for what other variables, as this was not detailed in the methods?

Reviewer #3: (No Response)

**Conclusions:**

-Are the conclusions supported by the data presented?

-Are the limitations of analysis clearly described?

-Do the authors discuss how these data can be helpful to advance our understanding of the topic under study?

-Is public health relevance addressed?

Reviewer #1: (No Response)

Reviewer #2: Public health relevance, strengths and limitations, were clearly discussed.

Reviewer #3: (No Response)

**Editorial and Data Presentation Modifications?**

Reviewer #1: (No Response)

Reviewer #2: Abstract, Line 48 and 52: “Fourteen districts had villages with an MHI prevalence48 of ≥2% for at least one STH species among SAC.” – SAC as the abbreviation for school-aged children will need to be written in full at first mention, same for AOR which I assume refers to adjusted odds ratio.

Line 148: ”…(500 mg) Samples were tested...”. Suggest having the manuscript checked for punctuation/language.

Line 157: “Schistosomiasis” can already be abbreviated here.

Line 184-185: “as positive” was mentioned twice in the same sentence.

Figure 5: Text is not clear especially within the graph, a clearer figure will be useful. The captions in grey font below the x-axis “Estimates are absolute differences between surveys…” should also be in the figure caption (Line 523).

Line 195: “Any STH status indicates the presence of any STH infection, including Ascaris lumbricoides, hookworm, and Trichuris trichiura”. Scientific names were not italicised here.

Reviewer #3: (No Response)

**Summary and General Comments:**

Reviewer #1: This paper reports on a national scale helminth survey in Rwanda, following several years of treatment. It’s timely as it compares infection with earlier surveys in 2008 and 2020. The sample size is large and involved the collections of duplicate Kato – Katz slides and a large data set examining risk factors for infection. There are a few areas where it could be strengthened and / or additional information provided before publication.

Comments

My biggest questions are around study design: in particular the geographical level at which results are calculated, and the sampling approach

• Geographical level of mapping / implementation. The WHO has moved towards recommending sub-district level mapping and implementation. In the Study design section (and similar in lines 699-723) the authors describe: ‘This study conducted a nationwide, community-based cross-sectional survey using a precision mapping approach to assess the prevalence and identify risk factors associated with SCH and STH.” And in lines 699-723 the authors talk about “A key strength of the study was the use of precision mapping, which provided a granular understanding of SCH, a focally distributed disease, resulting in more accurate prevalence estimates” and “The 2020 remapping exercise demonstrated how targeted sampling across diverse ecological settings can accurately identify transmission patterns at the community level”. But I only see results presented at the sector (district) level. Is that the level of implementation? To truly shrink the map then sub-district mapping and implementation is likely required. I do not think I consider the current approach as precision mapping, certainly not as most recently defined by WHO in the SPPA protocols. Was the mapping powered to give sub-district level results? If not, then I think much of section 699-723 needs amending.

• Sampling approach: In lines 294-295 the authors describe: “A three-stage sampling design was used to select study villages, focusing on areas likely to support schistosomiasis transmission due to their proximity to water bodies or wetlands.” (my emphasis). I think this is a reasonable programmatic approach when trying to (understandably) focus resources on the highest infection areas. But, by definition, it will lead to a biased estimate of sector / province / national level infection. And will inflate the current estimates and therefore underestimate any reductions since previous surveys (did they follow the same approach). I think this is worth the authors’ noting and discussing.

• The authors describe MHI infections as rare. MHI in Ascaris lumbricoides was 8.1%. This seems quite substantial to me, and more than would be expected from an overall Ascaris prevalence of 27.0%.

• How do the authors interpret the seeming conflicts in the SCH results: on one hand all sites had MHI < 1% so technically they have achieved elimination as a public health problem. But on the other hand, prevalence is still at 27% so continued MDA is required. Does this point to the need to update WHO guidance?

• The SCH prevalence levels (by KK) have hardly changed since 2008 (2.7% to 2.4%). What do the authors take from this?

• There is such a significant difference between SCH prevalence calculated by KK (1.7%) and CCA (27.2%). Even with trace negative it’s a 6-fold difference. What is the authors’ interpretation of this?

• The authors describe only 14 districts that had villages with >2% MHI? That seems very low when compared to overall MHI prevalence of Ascaris of 8.1%. Can it be checked?

• The blood in urine data are not presented. Was there no S. haematobium found? Was any urine filtration done?

• The survey was conducted in 2020. How much treatment has there been since then?

• Abstract – the authors do not mention overall SCH prevalence, as they do with STH. I suggest adding.

• Abstract – The odds ratios reported here, are they univariate or multivariate? What’s the theory behind being single being a risk factor?

• STH – Why do you think there is a drop in hookworm from 2008, but not in other STH species?

• Line 614-615: “This stagnation suggests that high MDA coverage in SAC alone may be insufficient to interrupt transmission.” Agreed, this is not a surprise.

• The option of combining IVM with ALB is well noted and worthy of further study.

Minor / editorial

• Lines 195-196, 425, 428 – species names should be in italics. Check throughout.

• Line 436 – repetition of ‘Additionally’.

Reviewer #2: The authors presented a nationwide remapping survey as an update to the last two surveys in 2008 and 2014, with the inclusion of adults and sampling of schools/villages in smaller administrative units. Compared to previous surveys focused upon school-aged children, this is the first community-based survey assessing prevalence across three targeted age groups: pre-school age, school age and adults. The findings highlight the need to include adults in MDA programs and improving access to clean water and sanitation for more sustainable disease control, noting the limited effectiveness of albendazole against Trichuriasis. Results were clearly described with maps to illustrate the data, however some methods require clarification.

Reviewer #3: (No Response)

PLOS authors have the option to publish the peer review history of their article (what does this mean? ). If published, this will include your full peer review and any attached files.

**Do you want your identity to be public for this peer review?** For information about this choice, including consent withdrawal, please see our Privacy Policy .

Reviewer #1: No

Reviewer #2: No

Reviewer #3: **Yes: ** Collins Okoyo

**Figure resubmission:**

**Reproducibility:**



---

## [Decision Letter · Decision Letter 1]

8 Jul 2025

Dear Dr. NSHIMIYIMANA,

We are pleased to inform you that your manuscript 'Remapping Parasite Landscapes: Nationwide Prevalence, Intensity and Risk Factors of Schistosomiasis and Soil-Transmitted Helminthiasis in Rwanda' has been provisionally accepted for publication in PLOS Neglected Tropical Diseases.

Best regards,

Angela Monica Ionica, Ph.D.

Academic Editor

Jong-Yil Chai

Section Editor

Shaden Kamhawi

co-Editor-in-Chief

Paul Brindley

co-Editor-in-Chief

Reviewer's Responses to Questions

**Key Review Criteria Required for Acceptance?**

**Methods**

-Are the objectives of the study clearly articulated with a clear testable hypothesis stated?

-Is the study design appropriate to address the stated objectives?

-Is the population clearly described and appropriate for the hypothesis being tested?

-Is the sample size sufficient to ensure adequate power to address the hypothesis being tested?

-Were correct statistical analysis used to support conclusions?

-Are there concerns about ethical or regulatory requirements being met?

Reviewer #3: The methods are well described in this revised version

**Results**

-Does the analysis presented match the analysis plan?

-Are the results clearly and completely presented?

-Are the figures (Tables, Images) of sufficient quality for clarity?

Reviewer #3: The results are well described in this revised version

**Conclusions**

-Are the conclusions supported by the data presented?

-Are the limitations of analysis clearly described?

-Do the authors discuss how these data can be helpful to advance our understanding of the topic under study?

-Is public health relevance addressed?

Reviewer #3: The conclusions are well stated

**Editorial and Data Presentation Modifications?**

Reviewer #3: (No Response)

**Summary and General Comments**

Reviewer #3: All my initial comments have been addressed, I don’t have further comments

PLOS authors have the option to publish the peer review history of their article (what does this mean? ). If published, this will include your full peer review and any attached files.

**Do you want your identity to be public for this peer review?** For information about this choice, including consent withdrawal, please see our Privacy Policy .

Reviewer #3: **Yes: ** Collins Okoyo

---

## [Editor Report · Acceptance letter]

Dear Mr. Nshimiyimana,

We are delighted to inform you that your manuscript, "Remapping Parasite Landscapes: Nationwide Prevalence, Intensity and Risk Factors of Schistosomiasis and Soil-Transmitted Helminthiasis in Rwanda," has been formally accepted for publication in PLOS Neglected Tropical Diseases.

Best regards,

Shaden Kamhawi

co-Editor-in-Chief

Paul Brindley

co-Editor-in-Chief
